# Deep Equilibrium Based Neural Operators for Steady-State PDEs

**Tanya Marwah**[1*]    **Ashwini Pokle** [1*]    **J. Zico Kolter** [1,2]    **Zachary C. Lipton** [1]
**Jianfeng Lu** [3]    **Andrej Risteski** [1]
[1]Carnegie Mellon University    [2] Bosch Center for AI    [3] Duke University
{tmarwah,apokle,zicokolter,zlipton,aristesk}@andrew.cmu.edu
jianfeng@math.duke.edu

## Abstract

Data-driven machine learning approaches are being increasingly used to solve partial differential equations (PDEs). They have shown particularly striking successes when training an operator, which takes as input a PDE in some family, and outputs its solution. However, the architectural design space, especially given structural knowledge of the PDE family of interest, is still poorly understood. We seek to remedy this gap by studying the benefits of weight-tied neural network architectures for steady-state PDEs. To achieve this, we first demonstrate that the solution of most steady-state PDEs can be expressed as a fixed point of a non-linear operator. Motivated by this observation, we propose FNO-DEQ, a deep equilibrium variant of the FNO architecture that directly solves for the solution of a steady-state PDE as the infinite-depth fixed point of an implicit operator layer using a black-box root solver and differentiates analytically through this fixed point resulting in $\mathcal{O}(1)$ training memory. Our experiments indicate that FNO-DEQ-based architectures outperform FNO-based baselines with $4\times$ the number of parameters in predicting the solution to steady-state PDEs such as Darcy Flow and steady-state incompressible Navier-Stokes. Finally, we show FNO-DEQ is more robust when trained with datasets with more noisy observations than the FNO-based baselines, demonstrating the benefits of using appropriate inductive biases in architectural design for different neural network based PDE solvers. Further, we show a universal approximation result that demonstrates that FNO-DEQ can approximate the solution to any steady-state PDE that can be written as a fixed point equation.

## 1 Introduction

Partial differential equations (PDEs) are used to model a wide range of processes in science and engineering. They define a relationship of (unknown) function and its partial derivatives. Most PDEs do not admit a closed form solution, and are solved using a variety of classical numerical methods such as finite element [LeVeque, 2007], finite volume [Moukalled et al., 2016], and spectral methods [Kopriva, 2009, Boyd, 2001]. These methods are often very computationally expensive, both as the ambient dimension grows, and as the desired accuracy increases.

This has motivated a rapidly growing area of research in data-driven approaches to PDE solving. One promising approach involves learning *neural solution operators* [Chen and Chen, 1995, Lu et al., 2019, Bhattacharya et al., 2021, Li et al., 2020b], which take in the coefficients of a PDE in some family and output its solution—and are trained by examples of coefficient-solution pairs.

---

*Equal contribution. Correspondence to tmarwah@andrew.cmu.edu and apokle@andrew.cmu.edu

37th Conference on Neural Information Processing Systems (NeurIPS 2023).

While several architectures for this task have been proposed, the design space—in particular taking into account structural properties of the PDEs the operator is trained on—is still largely unexplored. Most present architectures are based on "neuralizing" a classical numerical method. For instance, Li et al. [2020a] take inspiration from spectral methods, and introduce FNO: a trained composition of (parametrized) kernels in Fourier space. Brandstetter et al. [2022] instead consider finite-difference methods and generalize them into (learnable) graph neural networks using message-passing.

Our work focuses on families of PDEs that describe the steady-state of a system (that is, there is no time variable). Namely, we consider equations of the form:

$$L(a(x), u(x)) = f(x), \qquad \forall x \in \Omega, \tag{1}$$

where $u : \Omega \to \mathbb{R}^{d_u}$, $a : \Omega \to \mathbb{R}^{d_a}$ and $f : \Omega \to \mathbb{R}^{d_f}$ are functions defined over the domain $\Omega$, and $L$ is a (possibly non-linear) operator. This family includes many natural PDE families like Poisson equations, electrostatic equations, and steady-state Navier-Stokes.

We take inspiration from classical numerical approaches of fast-converging Newton-like iterative schemes [LeVeque, 2007, Faragó and Karátson, 2002] to solve steady-state PDEs, as well as recent theoretical works for elliptic (linear and non-linear PDEs) [Marwah et al., 2021, Chen et al., 2021, Marwah et al., 2022] to hypothesize that very deep, but heavily weight-tied architectures would provide a useful architectural design choice for steady-state PDEs.

In this paper, we show that for steady state equations it is often more beneficial to weight-tie an existing neural operator, as opposed to making the model deeper—thus increasing its size. To this end, we introduce **FNO-DEQ**, a new architecture for solving steady-state PDEs. FNO-DEQ is a deep equilibrium model (DEQ) that utilizes weight-tied FNO layers along with implicit differentiation and root-solvers to approximate the solution of a steady-state PDE. DEQs are a perfect match to the desiderata laid out above: they can be viewed alternately as directly parameterizing the fixed points of some iterative process; or by explicitly expanding some iterative fixed point solver like Newton's or Broyden's method as an infinitely deep, weight-tied model.

Such an architecture has a distinct computational advantage: implicit layer models effectively backpropagate through the infinite-depth network while using only constant memory (equivalent to a single layer's activations). Empirically, we show that for steady-state PDEs, weight-tied and DEQ based models perform better than baselines with $4\times$ the number of parameters, and are robust to training data noise. In summary, we make the following contributions:

- We show the benefits of weight-tying as an effective architectural choice for neural operators when applied to steady-state PDEs.

- We introduce FNO-DEQ, a FNO based deep equilibrium model (DEQ) that uses implicit layers and root solving to approximate the solution of a steady-state PDE. We further attest to the empirical performance of FNO-DEQ by showing that it performs as well as FNO and its variants with $4\times$ number of parameters.

- We show that FNO-DEQ and weight tied architectures are more robust to both input and observation noise, thus showing that weight-tying is a useful inductive bias for architectural design for steady-state PDEs.

- By leveraging the universal approximation results of FNO [Kovachki et al., 2021a] we show that FNO-DEQ based architectures can universally approximate the solution operator for a wide variety of steady-state PDE families.

- Finally, we create a dataset of pairs of steady-state incompressible Navier-Stokes equations with different forcing functions and viscosities, along with their solutions, which we will make public as a community benchmark for steady-state PDE solvers.

## 2   Related Work

Neural network based approaches for solving PDEs can broadly be divided into two categories. First are hybrid solvers [Bar-Sinai et al., 2019, Kochkov et al., 2021, Hsieh et al., 2019] which use neural networks in conjunction with existing numerical solvers. The main motivation is to not only improve upon the existing solvers, but to also replace the more computationally inefficient parts of the solver

with a learned counter part. Second set of approaches are full machine learning based approaches that aim to leverage the approximation capabilities of neural networks [Hornik et al., 1989] to directly learn the dynamics of the physical system from observations.

Hybrid solvers like Hsieh et al. [2019] use a neural network to learn a correction term to correct over an existing hand designed solver for a Poisson equation, and also provide convergence guarantees of their method to the solution of the PDE. However, the experiments in their paper are limited to linear elliptic PDEs. Further, solvers like Bar-Sinai et al. [2019] use neural networks to derive the discretizations for a given PDE, thus enabling the use of a low-resolution grid in the numerical solver. Furthermore, Kochkov et al. [2021] use neural networks to interpolate differential operators between grid points of a low-resolution grid with high accuracy. This work specifically focuses on solving Navier-Stokes equations, their method is more accurate than numerical techniques like Direct Numerical Simulation (DNS) with a low-resolution grid, and is also $80\times$ more faster. Brandstetter et al. [2022] introduced a message passing based hybrid scheme to train a hybrid solver and also propose a loss term which helps improve the stability of hybrid solvers for time dependent PDEs. However, most of these methods are equation specific, and are not easily transferable to other PDEs from the same family.

The neural network based approach that has recently garnered the most interest by the community is that of the operator learning framework [Chen and Chen, 1995, Kovachki et al., 2021b, Lu et al., 2019, Li et al., 2020a, Bhattacharya et al., 2021], which uses a neural network to approximate and infinite dimensional operator between two Banach spaces, thus learning an entire family of PDEs at once. Lu et al. [2019] introduces DeepONet, which uses two deep neural networks, referred to as the branch net and trunk net, which are trained concurrently to learn from data. Another line of operator learning framework is that of neural operators Kovachki et al. [2021b]. The most successful methodology for neural operators being the Fourier neural operators (FNO) [Li et al., 2020a]. FNO uses convolution based integral kernels which are evaluated in the Fourier space. Future works like Tran et al. [2021] introduce architectural improvements that enables one to train deeper FNO networks, thus increasing their size and improving their the performance on a variety of (time-dependent) PDEs. Moreover, the success of Transformers in domains like language and vision has also inspired transformer based neural operators in works like Li et al. [2022a], Hao et al. [2023] and Liu et al. [2022]. Theoretical results pertaining to the neural operators mostly include universal approximation results Kovachki et al. [2021a], Lanthaler et al. [2022] which show that architectures like FNO and DeepONet can indeed approximate the infinite dimension operators.

In this work, we focus on steady-state equations and show the benefits of weight-tying in improving the performance of FNO for steady-state equations. We show that instead of making a network deeper and hence increasing the size of a network, weight-tied FNO architectures can outperform FNO and its variants $4\times$ its size. We further introduce FNO-DEQ, a deep equilibrium based architecture to simulate an infinitely deep weight-tied network (by solving for a fixed point) with $\mathcal{O}(1)$ training memory. Our work takes inspiration from recent theoretical works like Marwah et al. [2021], Chen et al. [2021], Marwah et al. [2022] which derive parametric rates for some-steady state equations, and in fact prove that neural networks can approximate solutions to some families of PDEs with just poly$(d)$ parameters, thus evading the curse of dimensionality.

## 3 Preliminaries

We now introduce some key concepts and notation.

**Definition 1** ($L^2(\Omega; \mathbb{R}^d)$). *For a domain $\Omega$ we denote by $L^2(\Omega; \mathbb{R}^d)$ the space of square integrable functions $g : \Omega \to \mathbb{R}^d$ such that $\|g\|_{L^2(\Omega)} < \infty$, where $\|g\|_{L^2(\Omega)} = \left(\int_\Omega \|g(x)\|_{\ell_2}^2 dx\right)^{1/2}$.*

### 3.1 Neural Operators

Neural operators [Lu et al., 2019, Li et al., 2020a, Bhattacharya et al., 2021, Patel et al., 2021, Kovachki et al., 2023] are a deep learning approach to learning solution operators which map a PDE to its solution. Fourier Neural Operator (FNO) [Li et al., 2020a] is a particularly successful recent architecture parametrized as a sequence of kernel integral operator layers followed by non-linear activation functions. Each kernel integral operator layer is a convolution-based kernel function that is instantiated through a linear transformation in Fourier domain, making it less sensitive to the level of

spatial discretization. Specifically, an $L$-layered FNO $G_\theta : \mathbb{R}^{d_u} \to \mathbb{R}^{d_u}$ with learnable parameters $\theta$, is defined as

$$G_\theta := \mathcal{Q} \circ \mathcal{L}_L \circ \mathcal{L}_{L-1} \circ \cdots \circ \mathcal{L}_1 \circ \mathcal{P} \tag{2}$$

where $\mathcal{P} : L^2(\Omega; \mathbb{R}^{d_u}) \to L^2(\mathbb{R}^{d_v}; \mathbb{R}^{d_v})$ and $\mathcal{Q} : L^2(\mathbb{R}^{d_v}; \mathbb{R}^{d_v}) \to L^2(\mathbb{R}^{d_v}; \mathbb{R}^{d_u})$ are projection operators, and $\mathcal{L}_l : L^2(\mathbb{R}^{d_v}; \mathbb{R}^{d_v}) \to L^2(\mathbb{R}^{d_v}; \mathbb{R}^{d_v})$ for $l \in [L]$ is the $l^{\text{th}}$ FNO layer defined as,

$$\mathcal{L}_l(v_l) = \sigma\left(W_l v_l + b_l + \mathcal{K}_l(v_l)\right). \tag{3}$$

Here $\sigma$ is a non-linear activation function, $W_l, b_l$ are the $l^{th}$ layer weight matrix and bias terms. Finally $\mathcal{K}_l$ is the $l^{th}$ integral kernel operator which is calculated using the Fourier transform as introduced in Li et al. [2020a] defined as follows,

$$\mathcal{K}_l(v_l) = \mathcal{F}^{-1}\left(R_l \cdot (\mathcal{F} v_l)\right)(x) \qquad \forall x \in \Omega, \tag{4}$$

where $\mathcal{F}$ and $\mathcal{F}^{-1}$ are the Fourier transform and the inverse Fourier transform, with $R_l$ representing the learnable weight-matrix in the Fourier domain. Therefore, ultimately, the trainable parameters $\theta$ is a collection of all the weight matrices and biases, i.e, $\theta := \{W_l, b_l, R_l, \cdots, W_1, b_1, R_1\}$.

## 3.2  Equilibrium Models

Equilibrium models [Liao et al., 2018, Bai et al., 2019, Revay et al., 2020, Winston and Kolter, 2020] compute internal representations by solving for a fixed point in their forward pass. Specifically, consider a deep feedforward network with $L$ layers :

$$z^{[i+1]} = f_\theta^{[i]}\left(z^{[i]}; x\right) \quad \text{for } i = 0, ..., L-1 \tag{5}$$

where $x \in \mathbb{R}^{n_x}$ is the input injection, $z^{[i]} \in \mathbb{R}^{n_z}$ is the hidden state of $i^{th}$ layer with $z^{[0]} = \mathbf{0}$, and $f_\theta^{[i]} : \mathbb{R}^{n_x \times n_z} \mapsto \mathbb{R}^{n_z}$ is the feature transformation of $i^{th}$ layer, parametrized by $\theta$. Suppose the above model is weight-tied, *i.e.,* $f_\theta^{[i]} = f_\theta, \forall i$, and $\lim_{i \to \infty} f_\theta\left(z^{[i]}; x\right)$ exists and its value is $z^\star$. Further, assume that for this $z^\star$, it holds that $f_\theta\left(z^\star; x\right) = z^\star$. Then, equilibrium models can be interpreted as the infinite-depth limit of the above network such that $f_\theta^\infty\left(z^\star; x\right) = z^\star$

Under certain conditions[2], and for certain classes of $f_\theta$[3], the output $z^\star$ of the above weight-tied network is a fixed point. A simple way to solve for this fixed point is to use fixed point iterations, *i.e.,* repeatedly apply the update $z^{[t+1]} = f_\theta(z^{[t]}; x)$ some fixed number of times, and backpropagate through the network to compute gradients. However, this can be computationally expensive. Deep equilibrium (DEQ) models [Bai et al., 2019] explicitly solve for $z^\star$ through iterative root finding methods like Broyden's method [Broyden, 1965], Newton's method, Anderson acceleration [Anderson, 1965]. DEQs use implicit function theorem to directly differentiate through the fixed point $z^\star$ at equilibrium, thus requiring constant memory to backpropagate through an infinite-depth network:

$$\frac{\partial z^\star}{\partial \theta} = \left(I - \frac{\partial f_\theta(z^\star; x)}{\partial z^\star}\right)^{-1} \frac{\partial f_\theta(z^\star; x)}{\partial \theta} \tag{6}$$

Computing the inverse of Jacobian can quickly become intractable as we deal with high-dimensional feature maps. One can replace the inverse-Jacobian term with an identity matrix *i.e.,* Jacobian-free [Fung et al., 2022] or an approximate inverse-Jacobian [Geng et al., 2021] without affecting the final performance. There are alternate formulations of DEQs [Winston and Kolter, 2020] that guarantee existence of a unique equilibrium point. However, designing $f_\theta$ for these formulations can be challenging, and in this work we use the formulation by Bai et al. [2019].

## 4  Problem setting

We first formally define the system of steady-state PDEs that we will solve for:

---

[2]The fixed point can be reached if the dynamical system is globally contractive. This is usually not true in practice for most choices of $f_\theta$, and divergence is possible.

[3]Bai et al. [2019] state that $f_\theta$ needs to be stable and constrained. In general, by Banach's fixed point theorem, global convergence is guaranteed if $f_\theta$ is contractive over its input domain.

**Definition 2** (Steady-State PDE). *Given a bounded open set $\Omega \subset \mathbb{R}^d$, a steady-state PDE can be written in the following general form:*

$$L(a(x), u(x)) = f(x), \qquad \forall x \in \Omega \tag{7}$$

*Here $L$ is a continuous operator, the function $u \in L^2\left(\Omega; \mathbb{R}^{d_u}\right)$ is the unknown function that we wish to solve for and $a \in L^2\left(\Omega; \mathbb{R}^{d_a}\right)$ collects all the coefficients describing the PDE, and $f \in L^2\left(\Omega; \mathbb{R}^{d_f}\right)$ is a function independent of $u$. We will, for concreteness, assume periodic boundary conditions, i.e. $\forall z \in \mathbb{Z}^d, \forall x \in \Omega$ we have $u(x + z) = u(x)$. (Equivalently, $\Omega := \mathbb{T}^d = [0, 2\pi]^d$ can be identified with the torus.) [4] Finally, we will denote $u^\star : \Omega \to \mathbb{R}$ as the solution to the PDE.*

Steady-state models a system at stationarity, *i.e.,* when some quantity of interest like temperature or velocity no longer changes over time. Classical numerical solvers for these PDEs include iterative methods like Newton updates or conjugate gradient descent, typically with carefully chosen preconditioning to ensure benign conditioning and fast convergence. Furthermore, recent theoretical works [Marwah et al., 2021, Chen et al., 2021, Marwah et al., 2022] show that for many families of PDEs (e.g., steady-state elliptic PDEs that admit a variational formulation), iterative algorithms can be efficiently "neuralized", that is, the iterative algorithm can be represented by a compact neural network, so long as the coefficients of the PDE are also representable by a compact neural network. Moreover, the architectures constructed in these works are heavily weight-tied.

We will operationalize these developments through the additional observation that all these iterative schemes can be viewed as algorithms to find a fixed point of a suitably chosen operator. Namely, we can design an operator $\mathcal{G} : L^2(\Omega; \mathbb{R}^{d_u}) \times L^2(\Omega; \mathbb{R}^{d_f}) \to L^2(\Omega; \mathbb{R}^{d_u})$ [5] such that $u^\star = \mathcal{G}(u^\star, f)$ and a lot of common (preconditioned) first and second-order methods are natural ways to recover the fixed points $u^\star$.

Before describing our architectures, we introduce two components that we will repeatedly use.

**Definition 3** (Projection and embedding layers). *A projection and embedding layer, respectively $\mathcal{P} : L^2(\Omega; \mathbb{R}^{d_u}) \times L^2(\Omega; \mathbb{R}^{d_f}) \to L^2(\mathbb{R}^{d_v}; \mathbb{R}^{d_v}) \times L^2(\mathbb{R}^{d_v}; \mathbb{R}^{d_v})$ and $\mathcal{Q} : L^2(\mathbb{R}^{d_v}; \mathbb{R}^{d_v}) \to L^2(\mathbb{R}^{d_v}; \mathbb{R}^{d_u})$, are defined as*

$$\mathcal{P}(v, f) = \left(\sigma\left(W_P^{(1)} v + b_P^{(1)}\right), \sigma\left(W_P^{(2)} f + b_P^{(2)}\right)\right),$$
$$\mathcal{Q}(v) = \sigma\left(W_Q v + b_Q\right)$$

*where $W_P^{(1)} \in \mathbb{R}^{d_u \times d_v}, W_P^{(2)} \in \mathbb{R}^{d_f \times d_v}, W_Q \in \mathbb{R}^{d_v \times d_u}$ and $b_P^{(1)}, b_P^{(2)} \in \mathbb{R}^{d_v}, b_Q \in \mathbb{R}^{d_u}$.*

**Definition 4** (Input-injected FNO layer). *An input-injected FNO layer $\mathcal{L} : L^2(\mathbb{R}^{d_v}; \mathbb{R}^{d_v}) \times L^2(\mathbb{R}^{d_v}; \mathbb{R}^{d_v}) \to L^2(\mathbb{R}^{d_v}; \mathbb{R}^{d_v})$ is defined as*

$$\mathcal{L}(v, g) := g + \sigma\left(Wv + b + \mathcal{F}^{-1}(R^{(k)} \cdot (\mathcal{F}v))\right). \tag{8}$$

*where $W \in \mathbb{R}^{d_v \times d_v}$, $b \in \mathbb{R}^{d_v}$ and $R^{(k)} \in \mathbb{R}^{d_v \times d_v}$ for all $k \in [K]$ are learnable parameters.*

Note the difference between the FNO layer specified above, and the standard FNO layer Equation 3 is the extra input $g$ to the layer, which in our architecture will correspond to a projection of (some or all) of the PDE coefficients. We also note that this change to the FNO layer also enables us to learn deeper FNO architectures, as shown in Section 5. With this in mind, we can discuss the architectures we propose.

**Weight-tied architecture I: Weight-tied FNO** The first architecture we consider is a weight-tied version of FNO (introduced in Section 3), in which we repeatedly apply ($M$ times) the same transformation in each layer. More precisely, we have:

**Definition 5** (FNO Weight-Tied). *We define a $M$ times weight-tied neural operator $G_\theta^M$ as,*

$$G_\theta^M = \mathcal{Q} \circ \underbrace{\mathcal{B}^L \circ \mathcal{B}^L \circ \cdots \circ \mathcal{B}^L}_{M \text{ times}} \circ \mathcal{P}$$

---

[4]This is for convenience of exposition, our methods can readily be extended to other boundary conditions like Dirichet, Neumann etc.

[5]We note that the choice of defining the operator with the forcing function $f$ is made for purely expository purposes the operator $\mathcal{G}$ can be defined as $\mathcal{G} : L^2(\Omega; \mathbb{R}^{d_u}) \times L^2(\Omega; \mathbb{R}^{d_a}) \to L^2(\Omega; \mathbb{R}^{d_u})$ as well.

*such that:* $\mathcal{P}, \mathcal{Q}$ *are projection and embedding layers as in Definition 3 an* $\mathcal{B}^L : L^2(\mathbb{R}^{d_v}; \mathbb{R}^{d_v}) \times L^2(\mathbb{R}^{d_v}; \mathbb{R}^{d_v}) \to L^2(\mathbb{R}^{d_v}; \mathbb{R}^{d_v})$ *is an L-layer FNO block, i.e,* $\mathcal{B}^L = \mathcal{L}_L \circ \mathcal{L}_{L-1} \circ \mathcal{L}_{L-2} \circ \mathcal{L}_1$ *where for all* $l \in [L]$, $\mathcal{L}_l(\cdot, \mathcal{P}(f))$ [6] *is an input-injected FNO block as in Definition 4.*

**Weight-tied architecture II: FNO-DEQ** In cases where we believe a weight-tied $G_\theta^M$ converges to some fixed point as $M \to \infty$, unrolling $G_\theta^M$ for a large $M$ requires a lot of hardware memory for training: training the model requires one to store intermediate hidden units for each weight-tied layer for backpropagation, resulting in a $\mathcal{O}(M)$ increase in the amount of memory required.

To this end, we use Deep Equilibrium models (DEQs) which enables us to implicitly train $G_\theta := \lim_{M \to \infty} G_\theta^M$ by directly solving for the fixed point by leveraging black-box root finding algorithms like quasi-Newton methods, [Broyden, 1965, Anderson, 1965]. Therefore we can think of this approach as an infinite-depth (or infinitely unrolled) weight-tied network. We refer to this architecture as **FNO-DEQ**.

**Definition 6** (FNO-DEQ). *Given* $\mathcal{P}, \mathcal{Q}$ *and* $\mathcal{B}^L$ *in Definition 5, FNO-DEQ is trained to parametrize the fixed point equation* $\mathcal{B}^L(v^\star, \mathcal{P}(f)) = v^\star$ *and outputs* $u^\star = \mathcal{Q}(v^\star)$.

Usually, it is non-trivial to differentiate through these black-box root solvers. DEQs enable us to implicitly differentiate through the equilibrium fixed point efficiently without any need to backpropagate through these root solvers, therefore resulting in $\mathcal{O}(1)$ training memory.

# 5 Experiments

**Network architectures.** We consider the following network architectures in our experiments.

**FNO**: We closely follow the architecture proposed by Li et al. [2020a] and construct this network by stacking four FNO layers and four convolutional layers, separated by GELU activation [Hendrycks and Gimpel, 2016]. Note that in our current set up, we recover the original FNO architecture if the input to the $l^{\text{th}}$ layer is the output of $(l-1)^{\text{th}}$ layer *i.e.,* $v_l = \mathcal{B}_{l-1}(v_{l-1})$.

**Improved FNO (FNO++ )**: The original FNO architecture suffers from vanishing gradients, which prohibits it from being made deeper [Tran et al., 2021]. We overcome this limitation by introducing residual connections within each block of FNO, with each FNO block $\mathcal{B}_l$ comprising of three FNO layers $\mathcal{L}$ *i.e.,* $\mathcal{B}_l = \mathcal{L}_{L_1}^l \circ \mathcal{L}_{L_2}^l \circ \mathcal{L}_{L_3}^l$ and three convolutional layers, where $\mathcal{L}$ is defined in Eq. (8).

**Weight-tied network (FNO-WT)**: This is the weight-tied architecture introduced in Definition 5, where we initialize $v_0(x) = 0$ for all $x \in \Omega$.

**FNO-DEQ**: As introduced in Definition 6, FNO-DEQ is a weight-tied network where we explicitly solve for the fixed point in the forward pass with a root finding algorithm. We use Anderson acceleration [Anderson, 1965] as the root solver. For the backward pass, we use approximate implicit gradients [Geng et al., 2021] which are light-weight and more stable in practice, compared to implicit gradients computed by inverting Jacobian.

Note that both weight-tied networks and FNO-DEQs leverage weight-tying but the two models differ in the ultimate goal of the forward pass: DEQs explicitly solve for the fixed point during the forward pass, while weight-tied networks trained with backpropagation may or may-not reach a fixed point [Anil et al., 2022]. Furthermore, DEQs require $\mathcal{O}(1)$ memory, as they differentiate through the fixed point implicitly, whereas weight-tied networks need to explicitly create the entire computation graph for backpropagation, which can become very large as the network depth increases. Additionally, FNO++ serves as a non weight-tied counterpart to a weight-tied input-injected network. Like weight-tied networks, FNO++ does not aim to solve for a fixed point in the forward pass.

**Experimental setup.** We test the aforementioned network architectures on two families of steady-state PDEs: Darcy Flow equation and steady-state Navier-Stokes equation for incompressible fluids. For experiments with Darcy Flow, we use the dataset provided by Li et al. [2020a], and generate our own dataset for steady-state Navier-Stokes. For more details on the datasets and the data generation processes we refer to Sections B.1 and B.2 of the Appendix. For each family of PDE, we train networks under 3 different training setups: clean data, noisy inputs and noisy observations. For

---

[6] We are abusing the notation somewhat and denoting by $\mathcal{P}(f)$ the second coordinate of $\mathcal{P}$, which only depends on $f$.

experiments with noisy data, both input and observations, we add noise sampled from a sequence of standard Gaussians with increasing values of variance $\{\mathcal{N}(0, (\sigma_k^2))\}_{k=0}^{M-1}$, where $M$ is the total number of Gaussians we sample from. We set $\sigma_0^2 = 0$ and $\sigma_{\max}^2 = \sigma_{M-1}^2 \leq 1/r$, where $r$ is the resolution of the grid. Thus, the training data includes equal number of PDEs with different levels of Gaussian noise added to their input or observations. We add noise to training data, and always test on clean data. We follow prior work [Li et al., 2020b] and report relative $L_2$ norm between ground truth $u^\star$ and prediction on test data. The total depth of all networks besides FNO is given by $6B + 4$, where $B$ is the number of FNO blocks. Each FNO block has 3 FNO layers and convolutional layers. In addition, we include the depth due to $\mathcal{P}$, $\mathcal{Q}$, and an additional final FNO layer and a convolutional layer. We further elaborate upon network architectures and loss functions in in Appendix A.

## 5.1 Darcy Flow

For our first set of experiments we consider stationary Darcy Flow equations, a form of linear elliptic PDE with in two dimensions. The PDE is defined as follows,

$$
\begin{aligned}
-\nabla \cdot (a(x)\nabla u(x)) &= f(x), & x &\in (0,1)^2 \\
u(x) &= 0 & x &\in \partial(0,1)^2.
\end{aligned}
\tag{9}
$$

Note that the diffusion coefficient $a \in L^\infty(\Omega)(\Omega; \mathbb{R}_+)$, i.e., the coefficients are always positive, and $f \in L^2(\Omega; \mathbb{R}^{d_f})$ is the forcing term. These PDEs are used to model the steady-state pressure of fluids flowing through a porous media. They can also be used to model the stationary state of the diffusive process with $u(x)$ modeling the temperature distribution through the space with $a$ defining the thermal conductivity of the medium. The task is to learn an operator $G_\theta : L^2(\Omega; \mathbb{R}^{d_u}) \times L^2(\Omega; \mathbb{R}^{d_a}) \rightarrow L^2(\Omega; \mathbb{R}^{d_u})$ such that $u^\star = G_\theta(u^\star, a)$.

We report the results of our experiments on Darcy Flow in Table 1. The original FNO architecture does not improve its performance with increased number of FNO blocks $\mathcal{B}$. FNO++ with residual connections scales better but saturates at around 4 FNO blocks. In contrast, FNO-WT and FNO-DEQ with just a *single* FNO block outperform deeper non-weight-tied architectures on clean data and on data with noisy inputs. When observations are noisy, FNO-WT and FNO-DEQ outperform FNO++ with a similar number of parameters, and perform comparably to FNO++ with $4\times$ parameters.

We also report results on shallow FNO-DEQ, FNO-WT and FNO++ architectures. An FNO block in these shallow networks has a single FNO layer instead of three layers. In our experiments, shallow weight-tied networks outperform non-weight-tied architectures including FNO++ with $7\times$ parameters on clean data and on data with noisy inputs, and perform comparably to deep FNO++ on noisy observations. In case of noisy observations, we encounter training instability issues in FNO-DEQ. We believe that this shallow network lacks sufficient representation power and cannot accurately solve for the fixed point during the forward pass. These errors in fixed point estimation accumulate over time, leading to incorrect values of implicit gradients, which in turn result in training instability issues.

## 5.2 Steady-state Navier-Stokes Equations for Incompressible Flow

We consider the steady-state Navier-Stokes equation for an incompressible viscous fluid in the vorticity form defined on a torus, i.e., with periodic boundary condition,

$$
\begin{aligned}
u \cdot \nabla \omega &= \nu \Delta \omega + f, & x &\in \Omega \\
\nabla \cdot u &= 0 & x &\in \Omega
\end{aligned}
\tag{10}
$$

where $\Omega := (0, 2\pi)^2$, and $u : \Omega \rightarrow \mathbb{R}^2$ is the velocity and $\omega : \Omega \rightarrow \mathbb{R}$ where $\omega = \nabla \times u$, $\nu \in \mathbb{R}_+$ is the viscosity and $f : \Omega \rightarrow \mathbb{R}$ is the external force term. We learn an operator $G_\theta : L^2(\Omega; \mathbb{R}^{d_u}) \times L^2(\Omega; \mathbb{R}^{d_f}) \rightarrow L^2(\Omega; \mathbb{R}^{d_u})$, such that $u^\star = G_\theta(u^\star, f)$. We train all the models on data with viscosity values $\nu = 0.01$ and $\nu = 0.001$, and create a dataset for steady-state incompressible Navier-Stokes, which we will make public as a community benchmark for steady-state PDE solvers.

Results for Navier-Stokes equation have been reported in Table 2 and Table 3. For both values of viscosity, FNO-DEQ outperforms other architectures for all three cases: clean data, noisy inputs and noisy observations. FNO-DEQ is more robust to noisy inputs compared to non-weight-tied architectures. For noisy inputs, FNO-DEQ matches the test-error of noiseless case in case of viscosity 0.01 and almost matches the test-error of noiseless case in case of viscosity 0.001. We provide additional results for noise level 0.004 in Appendix E. FNO-DEQ and FNO-WT consistently outperform non-weight-tied architectures for higher levels of noise as well.

| Architecture | Parameters | #Blocks | Test error ↓ | | |
|---|---|---|---|---|---|
| | | | $\sigma_{\max}^2 = 0$ | $(\sigma_{\max}^2)^i = 0.001$ | $(\sigma_{\max}^2)^t = 0.001$ |
| FNO | 2.37M | 1 | $0.0080 \pm 5\text{e-}4$ | $0.0079 \pm 2\text{e-}4$ | $0.0125 \pm 4\text{e-}5$ |
| FNO | 4.15M | 2 | $0.0105 \pm 6\text{e-}4$ | $0.0106 \pm 4\text{e-}4$ | $0.0136 \pm 2\text{e-}5$ |
| FNO | 7.71M | 4 | $0.2550 \pm 2\text{e-}8$ | $0.2557 \pm 8\text{e-}9$ | $0.2617 \pm 2\text{e-}9$ |
| FNO++ | 2.37M | 1 | $0.0075 \pm 2\text{e-}4$ | $0.0075 \pm 2\text{e-}4$ | $0.0145 \pm 7\text{e-}4$ |
| FNO++ | 4.15M | 2 | $0.0065 \pm 2\text{e-}4$ | $0.0065 \pm 9\text{e-}5$ | $0.0117 \pm 5\text{e-}5$ |
| FNO++ | 7.71M | 4 | $0.0064 \pm 2\text{e-}4$ | $0.0064 \pm 2\text{e-}4$ | $\mathbf{0.0109 \pm 5\text{e-}4}$ |
| S-FNO++ | 1.78M | 0.66 | $0.0093 \pm 5\text{e-}4$ | $0.0094 \pm 7\text{e-}4$ | $0.0402 \pm 6\text{e-}3$ |
| FNO-WT | 2.37M | 1 | $\mathbf{0.0055 \pm 1\text{e-}4}$ | $\mathbf{0.0056 \pm 5\text{e-}5}$ | $0.0112 \pm 4\text{e-}4$ |
| FNO-DEQ | 2.37M | 1 | $\mathbf{0.0055 \pm 1\text{e-}4}$ | $\mathbf{0.0056 \pm 7\text{e-}5}$ | $0.0112 \pm 4\text{e-}4$ |
| S-FNO-WT | 1.19M | 0.33 | $0.0057 \pm 3\text{e-}5$ | $0.0057 \pm 5\text{e-}5$ | $0.0112 \pm 1\text{e-}4$ |
| S-FNO-DEQ | 1.19M | 0.33 | $0.0056 \pm 4\text{e-}5$ | $0.0056 \pm 5\text{e-}5$ | $0.0136 \pm 0.011$ |

Table 1: Results on Darcy flow: clean data (Col 4),noisy inputs (Col 5) and noisy observations (Col 6) with max variance of added noise being $(\sigma_{\max}^2)^i$ and $(\sigma_{\max}^2)^t$, respectively. Reported test error has been averaged on three different runs with seeds 0, 1, and 2. Here, S-FNO++, S-FNO-WT and S-FNO-DEQ are shallow versions of FNO++, FNO-WT and FNO-DEQ respectively.

In general, DEQ-based architectures are slower to train (upto $\sim 2.5\times$ compared to feedforward networks of similar size) as we solve for the fixed point in the forward pass. However, their inductive bias provides performance gains that cannot be achieved by simply stacking non-weight-tied FNO layers. In general, we observe diminishing returns in FNO++ beyond 4 blocks. Additionally, training the original FNO network on more than 4 FNO blocks is challenging, with the network often diverging during training, and therefore we do not include these results in the paper.

| Architecture | Parameters | #Blocks | Test error ↓ | | |
|---|---|---|---|---|---|
| | | | $\sigma_{\max}^2 = 0$ | $(\sigma_{\max}^2)^i = 0.001$ | $(\sigma_{\max}^2)^t = 0.001$ |
| FNO | 2.37M | 1 | $0.184 \pm 0.002$ | $0.218 \pm 0.003$ | $0.184 \pm 0.001$ |
| FNO | 4.15M | 2 | $0.162 \pm 0.024$ | $0.176 \pm 0.004$ | $0.152 \pm 0.005$ |
| FNO | 7.71M | 4 | $0.157 \pm 0.012$ | $0.187 \pm 0.004$ | $0.166 \pm 0.013$ |
| FNO++ | 2.37M | 1 | $0.199 \pm 0.001$ | $0.230 \pm 0.001$ | $0.197 \pm 0.001$ |
| FNO++ | 4.15M | 2 | $0.154 \pm 0.005$ | $0.173 \pm 0.003$ | $0.154 \pm 0.006$ |
| FNO++ | 7.71M | 4 | $0.151 \pm 0.003$ | $0.165 \pm 0.004$ | $0.149 \pm 0.003$ |
| FNO-WT | 2.37M | 1 | $0.151 \pm 0.007$ | $0.173 \pm 0.017$ | $\mathbf{0.126 \pm 0.027}$ |
| FNO-DEQ | 2.37M | 1 | $\mathbf{0.128 \pm 0.004}$ | $\mathbf{0.144 \pm 0.007}$ | $0.136 \pm 0.003$ |

Table 2: Results on incompressible steady-state Navier-Stokes (viscosity=0.001): clean data (Col 4), noisy inputs (Col 5) and noisy observations (Col 6) with max variance of added noise being $(\sigma_{\max}^2)^i$ and $(\sigma_{\max}^2)^t$, respectively. Reported test error has been averaged on three different runs with seeds 0, 1, and 2.

## 6  Universal Approximation and Fast Convergence of FNO-DEQ

Though the primary contribution of our paper is empirical, we show (by fairly standard techniques) that FNO-DEQ is a universal approximator, under mild conditions on the steady-state PDEs. Moreover, we also show that in some cases, we can hope the fixed-point solver can converge rapidly.

As noted in Definition 2, we have $\Omega := \mathbb{T}^d$. We note that all continuous function $f \in L^2(\Omega; \mathbb{R})$ and $\int_\Omega |f(x)| dx < \infty$ can be written as, $f(x) = \sum_{\omega \in \mathbb{Z}^d} e^{ix^T \omega} \hat{f}_w$. where $\{\hat{f}_\omega\}_{\omega \in \mathbb{Z}^d}$ are the Fourier coefficients of the function $f$. We define as $L_N^2(\Omega)$ as the space of functions such that for all $f_N \in L_N^2(\Omega)$ with Fourier coefficients that vanish outside a bounded ball. Finally, we define an

| Architecture | Parameters | #Blocks | Test error ↓ | | |
|---|---|---|---|---|---|
| | | | $\sigma_{\max}^2 = 0$ | $(\sigma_{\max}^2)^i = 0.001$ | $(\sigma_{\max}^2)^t = 0.001$ |
| FNO | 2.37M | 1 | $0.181 \pm 0.005$ | $0.186 \pm 0.003$ | $0.178 \pm 0.006$ |
| FNO | 4.15M | 2 | $0.138 \pm 0.007$ | $0.150 \pm 0.006$ | $0.137 \pm 0.012$ |
| FNO | 7.71M | 4 | $0.152 \pm 0.006$ | $0.163 \pm 0.002$ | $0.151 \pm 0.008$ |
| FNO++ | 2.37M | 1 | $0.188 \pm 0.002$ | $0.207 \pm 0.004$ | $0.187 \pm 0.003$ |
| FNO++ | 4.15M | 2 | $0.139 \pm 0.004$ | $0.153 \pm 0.002$ | $0.140 \pm 0.005$ |
| FNO++ | 7.71M | 4 | $0.130 \pm 0.005$ | $0.151 \pm 0.004$ | $0.128 \pm 0.009$ |
| FNO-WT | 2.37M | 1 | $0.099 \pm 0.007$ | $0.101 \pm 0.007$ | $0.130 \pm 0.044$ |
| FNO-DEQ | 2.37M | 1 | $\mathbf{0.088 \pm 0.006}$ | $\mathbf{0.099 \pm 0.007}$ | $\mathbf{0.116 \pm 0.011}$ |

Table 3: Results on incompressible steady-state Navier-Stokes (viscosity=0.01): clean data (Col 4), noisy inputs (Col 5) and noisy observations (Col 6) with max variance of added noise being $(\sigma_{\max}^2)^i$ and $(\sigma_{\max}^2)^t$, respectively. Reported test error has been averaged on three different runs with seeds 0, 1, and 2.

orthogonal projection operator $\Pi_N : L^2(\Omega) \to L_N^2(\Omega)$, such that for all $f \in L^2(\Omega)$ we have,

$$f_n = \Pi_N(f) = \Pi_N \left( \sum_{\omega \in \mathbb{Z}^d} f_\omega e^{ix^T \omega} \right) = \sum_{\|\omega\|_\infty \leq N} \hat{f}_\omega e^{ix^T \omega}. \tag{11}$$

That is, the projection operator $\Pi_N$ takes an infinite dimensional function and projects it to a finite dimensional space. We prove the following universal approximation result:

**Theorem 1.** *Let $u^\star \in L^2(\Omega; \mathbb{R}^{d_u})$ define the solution to a steady-state PDE in Definition 2, Then there exists an operator $\mathcal{G} : L^2(\Omega; \mathbb{R}^{d_u}) \times L^2(\Omega; \mathbb{R}^{d_f}) \to L^2(\Omega; \mathbb{R}^{d_u})$ such that, $u^\star = \mathcal{G}(u^\star, f)$. Furthermore, for every $\epsilon > 0$ there exists an $N \in \mathbb{N}$ such that for compact sets $K_u \subset L^2(\Omega; \mathbb{R}^{d_u})$ and $K_f \subset L^2(\Omega; \mathbb{R}^{d_f})$ there exists a neural network $G_\theta : L_N^2(\Omega; \mathbb{R}^{d_u}) \times L_N^2(\Omega; \mathbb{R}^{d_f}) \to L_N^2(\Omega; \mathbb{R}^{d_u})$ with parameters $\theta$, such that,*

$$\sup_{u \in K_u, f \in K_f} \|u^\star - G_\theta(\Pi_N u^\star, \Pi_N f)\|_{L^2(\Omega)} \leq \epsilon.$$

The proof for the above theorem is relatively straightforward and provided in Appendix C. The proof uses the fact that $u^\star$ is a fixed-point of the operator $G(u, f) = u - (L(u) - f)$, allowing us to use the the results in Kovachki et al. [2021a] that show a continuous operator can be approximated by a network as defined in Equation 2. Note that the choice of $G$ is by no means unique: one can "universally approximate" any operator $G(u, f) = u - A(L(u) - f)$, for a continuous operator $A$. Such a $G$ can be thought of as a form of "preconditioned" gradient descent, for a preconditioner $A$. For example, a Newton update has the form $G(u, f) = u - L'(u)^{-1}(L(u) - f)$, where $L' : L^2(\Omega; \mathbb{R}^{d_u}) \to L^2(\Omega; \mathbb{R}^{d_u})$ is the Frechet derivative of the operator $L$.

The reason this is relevant is that the DEQ can choose to universally approximate a fixed-point equation for which the fixed-point solver it is trained with also converges rapidly. As an example, the following classical result shows that under Lax-Milgram-like conditions (a kind of strong convexity condition), Newton's method converges doubly exponentially fast:

**Lemma 1** (Faragó and Karátson [2002], Chapter 5). *Consider the PDE defined Definition 2, such that $d_u = d_v = d_f = 1$. such that $L'(u)$ defines the Frechet derivative of the operator $L$. If for all $u, v \in L^2(\Omega; \mathbb{R})$ we have $\|L'(u)v\|_{L^2(\Omega)} \geq \lambda \|v\|_{L^2(\Omega)}$ and $\|L'(u) - L'(v)\|_{L^2(\Omega)} \leq \Lambda \|u - v\|_{L^2(\Omega)}$ for $0 < \lambda \leq \Lambda < \infty$, then for the Newton update, $u_{t+1} \leftarrow u_t - L'(u_t)^{-1}(L(u_t) - f)$, with $u_0 \in L^2(\Omega; \mathbb{R})$, there exists an $\epsilon > 0$, such that $\|u_T - u^\star\|_{L^2(\Omega)} \leq \epsilon$ if $T \geq \log \left( \log \left( \frac{1}{\epsilon} \right) / \log \left( \frac{2\lambda^2}{\Lambda \|L(u_0) - f\|_{L^2(\Omega)}} \right) \right)$.*

For completeness, we include the proof of the above lemma in the Appendix (Section D). We note that the conditions of the above lemma are satisfied for elliptic PDEs like Darcy Flow, as well as many variational non-linear elliptic PDEs (e.g., those considered in Marwah et al. [2022]). Hence, we can expect FNO-DEQs to quickly converge to the fixed point, since they employ quasi-Newton methods like Broyden and Anderson methods [Broyden, 1965, Anderson, 1965].

# 7 Conclusion

In this work, we demonstrate that the inductive bias of deep equilibrium models—and weight-tied networks in general—makes them ideal architectures for approximating neural operators for steady-state PDEs. Our experiments on steady-state Navier-Stokes equation and Darcy flow equations show that weight-tied models and FNO-DEQ perform outperform FNO models with $\sim 4\times$ the number of parameters and depth. Our findings indicate that FNO-DEQ and weight-tied architectures are, in general, more robust to both input and observation noise compared to non-weight-tied architectures, including FNO. We believe that our results complement any future progress in the design and development of PDE solvers [Tran et al., 2021, Li et al., 2022b] for steady-state PDEs, and hope that our work motivates the study of relevant inductive biases that could be used to improve them.

# 8 Acknowledgements

TM is supported by CMU Software Engineering Institute via Department of Defense under contract FA8702-15-D-0002. AP is supported by a grant from the Bosch Center for Artificial Intelligence. ZL gratefully acknowledges the NSF (FAI 2040929 and IIS2211955), UPMC, Highmark Health, Abridge, Ford Research, Mozilla, the PwC Center, Amazon AI, JP Morgan Chase, the Block Center, the Center for Machine Learning and Health, and the CMU Software Engineering Institute (SEI) via Department of Defense contract FA8702-15-D-0002, for their generous support of ACMI Lab's research. JL is supported in part by NSF award DMS-2012286, and AR is supported in part by NSF awards IIS-2211907, CCF-2238523, Amazon Research Award, and the CMU/PwC DT&I Center.

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

# Appendix

## A   Implementation Details

**Training details.**   We train all the networks for 500 epochs with Adam optimizer. The learning rate is set to 0.001 for Darcy flow and 0.005 for Navier-Stokes. We use learning rate weight decay of 1e-4 for both Navier-Stokes and Darcy flow. The batch size is set to 32. In case of Darcy flow, we also use cosine annealing for learning rate scheduling. We run all our experiments on a combination of NVIDIA RTX A6000, NVIDIA GeForce RTX 2080 Ti and 3080 Ti. All networks can easily fit on a single NVIDIA RTX A6000, but training time varies between the networks.

For FNO-DEQ, we use Anderson solver [Anderson, 1965] to solve for the fixed point in the forward pass. The maximum number of Anderson solver steps is kept fixed at 32 for Dary Flow, and 16 for Navier Stokes. For the backward pass, we use phantom gradients [Geng et al., 2021] which are computed as:

$$u^\star = \tau G_\theta(u^\star, a) + (1 - \tau)u^\star \tag{12}$$

where $\tau$ is a tunable damping factor and $u^\star$ is the fixed point computed using Anderson solver in the forward pass. This step can be repeated $S$ times. We use $\tau = 0.5$ and $S = 1$ for Darcy Flow, and $\tau = 0.8$ and $S = 3$ for Navier-Stokes.

For the S-FNO-DEQ used in Table 1, we use Broyden's method [Broyden, 1965] to solve for the fixed point in the forward pass and use exact implicit gradients, computed through implicit function theorem as shown in Eq. (6), for the backward pass through DEQ. The maximum number of solver steps is fixed at 32.

For weight-tied networks, we repeatedly apply the FNO block to the input 12 times for Darcy flow, and 6 times for Navier-Stokes.

**Network architecture details.**   The width of an FNO layer set to 32 across all the networks. Additionally, we retain only 12 Fourier modes in FNO layer, and truncate higher Fourier modes. We use the code provided by Li et al. [2020a] to replicate the results for FNO, and construct rest of the networks on top of this as described in Sec. 5.

## B   Datasets

### B.1   Darcy Flow

As mentioned in Sec. 5 we use the dataset provided by Li et al. [2020a] for our experiments with steady-state Darcy-Flow.

All the models are trained on 1024 data samples and tested on 500 samples. The resolution of original images is $421 \times 421$ which we downsample to $85 \times 85$ for our experiments. For experiments with noisy inputs/observations, the variance of Gaussian noise that we add to PDEs are [0, 1e-9, 1e-8, 1e-7, 1e-6, 1e-5, 1e-4, 1e-3].

### B.2   Steady-State Incompressible Fluid Navier-Stoke

$$u \cdot \nabla \omega = \nu \Delta \omega + f, \qquad x \in \Omega$$
$$\nabla \cdot u = 0 \qquad\qquad x \in \Omega$$

To generate the dataset for steady-state Navier-Stokes, instead of solving the steady state PDE using steady-state solvers like the SIMPLE algorithm Patankar and Spalding [1983], we first choose the solution $\omega^\star := \nabla \times u^\star$ of the PDE and then generate the corresponding equation, i.e. calculate the corresponding force term $f = u^\star \cdot \nabla \omega^\star - \nu \Delta \omega^\star$.

To generate the solutions $\omega^\star$, we forward propagate a relatively simple initial distribution of $\omega_0$ (sampled from a Gaussian random field) through a time-dependent Navier-Stokes equation in the vorticity form for a short period of time. This ensures our dataset contains solutions $\omega^*$ that are rich

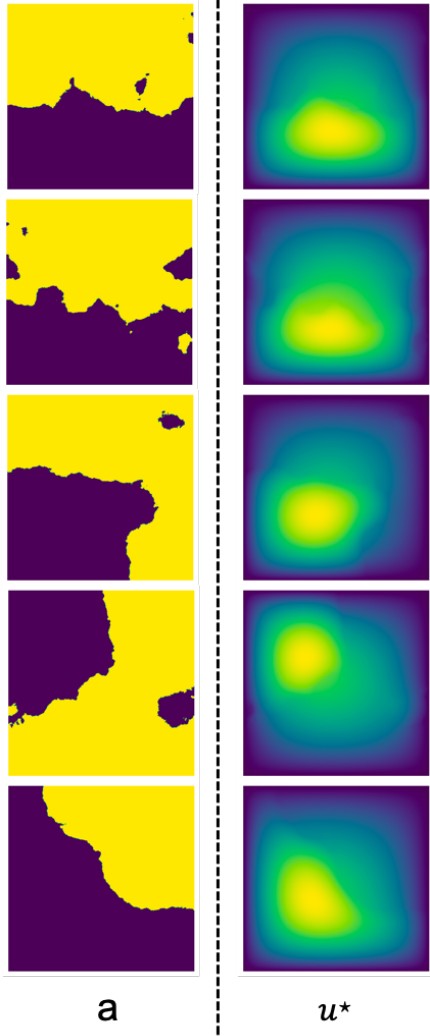

**a**           $u^{\star}$

Figure 1: Samples from Darcy Flow

and complex. Precisely, recall the Navier-Stokes equations in their vorticity form:

$$\partial_t \omega(x,t) + u(x,t) \cdot \nabla \omega(x,t) = \nu \Delta \omega(x,t) + g(x) \qquad x \in (0, 2\pi)^2, t \in [0, T]$$
$$\nabla \cdot u(x,t) = 0 \qquad x \in (0, 2\pi)^2, t \in [0, T] \qquad (13)$$
$$\omega(x,0) = \omega_0(x) \qquad x \in (0, 2\pi)^2$$

where $g(x) = \nabla \times \tilde{g}(x)$ and $\tilde{g}(x) = \sin(5x_1)\hat{x}_2$ is a divergence free forcing term and $x = (x_1, x_2)$ are the two coordinates of the input vector. We forward propagate the equations Equation 13 using a pseudo-spectral method using the functions provided in JAX-CFD [Kochkov et al., 2021, Dresdner et al., 2022] package. The initial vorticity $\omega_0$ is sampled from a Gaussian random field $\mathcal{N}(0, (5^{3/2}(I + 25\Delta)^{-2.5}))$, which is then made divergence free. We forward propagate the Navier-Stokes equation in Equation 13 for time $T = 0.5$ with $dt = 0.002$ to get $\omega(1, x)$, which we choose as the solution to the steady-state PDE in Equation 10, i.e, $\omega^{\star}$ for Equation 10.

Subsequently, we use the stream function $\Psi$ [Batchelor and Batchelor, 1967] to calculate $u = (\partial \Psi / \partial x_1, \partial \Psi / \partial x_2)$ by solving the Poisson equation $\Delta \Psi = \omega$ in the Fourier domain. Furthermore, since $f = u^{\star} \cdot \nabla \omega^{\star} - \nu \Delta \omega^{\star}$ we use the stream function to calculate $(f_1, f_2)$, i.e., the different components of the force term.

We use 4500 training samples and 500 testing samples. The input to the network is the vector field $\tilde{f} = (f_1, f_2)$ and we learn a map that outputs the vorticity $\omega^{\star}$. The resolution of grid used to

generate the dataset is $256 \times 256$ which we downsample to $128 \times 128$ while training the models. For experiments with noisy inputs/observations, we consider two values of maximum variance of Gaussian noise: 1e-3 and 4e-3. The variances of the Gaussian noise that we add to the PDEs for the latter case are [0, 1e-9, 1e-8, 1e-7, 1e-6, 1e-5, 1e-4, 1e-3, 2e-3, 4e-3]. However, when conducting experiments with a variance of 1e-3, we exclude the last two values of variance from this list.

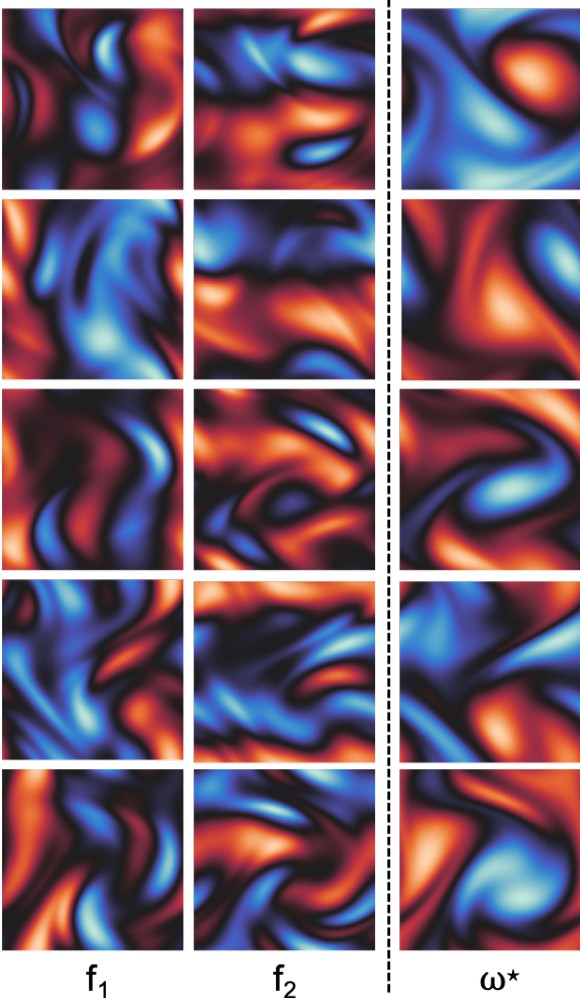

$$f_1 \qquad f_2 \qquad \omega^\star$$

Figure 2: Samples from Steady-state Navier-Stokes dataset with viscosity 0.001. Each triplet visualizes the inputs $f_1$, $f_2$ and the ground truth output i.e. $\omega^\star$.

## C Proof of Universal Approximation

The proof of the universal approximation essentially follows from the result on the universal approximation capabilities of FNO layers in Kovachki et al. [2021a], applied to $\mathcal{G}(v, f) = v - (Lv - f)$. For the sake of completeness, we reitarate the key steps.

For simplicity, we will assume that $d_u = d_v = d_f = 1$. (The results straightforwardly generalize.) We will first establish some key technical lemmas and introduce some notation and definitions useful for the proof for Theorem 1.

**Definition 7.** *An operator $T : L^2(\Omega; \mathbb{R}) \to L^2(\Omega; \mathbb{R})$ is continuous at $u \in L^2(\Omega; \mathbb{R})$ if for every $\epsilon > 0$, there exists a $\delta > 0$, such that for all $v \in L^2(\Omega)$ with $\|u - v\|_{L^2(\Omega)} \leq \delta$, we have $\|L(u) - L(v)\|_{L^2(\Omega)} \leq \epsilon$.*

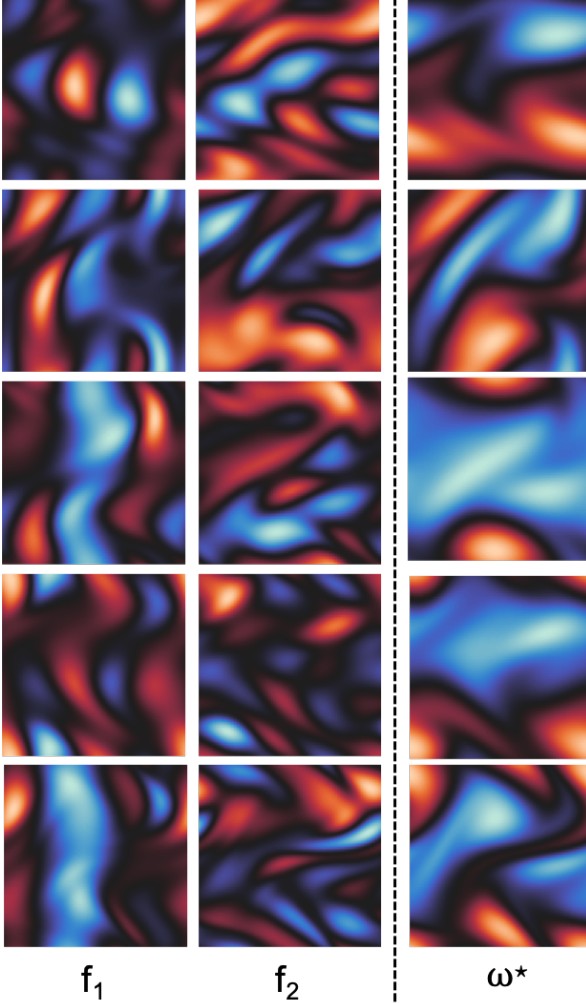

$$f_1 \qquad\qquad f_2 \qquad\qquad \omega^\star$$

Figure 3: Samples from Steady-state Navier-Stokes dataset with viscosity 0.01. Each triplet visualizes the inputs $f_1$, $f_2$ and the ground truth output i.e. $\omega^\star$.

First, we approximate the infinite dimensional operator $\mathcal{G} : L^2(\Omega) \times L^2(\Omega) \to L^2(\Omega)$ by projecting the functions in $L^2(\Omega)$ to a finite-dimensional approximation $L_N^2(\Omega)$, and considering the action of the operator on this subspace. The linear projection we use is the one introduced in Equation 11. More precisely we show the following result,

**Lemma 2.** *Given a continuous operator $L : L^2(\Omega) \to L^2(\Omega)$ as defined in Equation 1, let us define an operator $\mathcal{G} : L^2(\Omega) \times L^2(\Omega) \to L^2(\Omega)$ as $\mathcal{G}(v, f) := v - (L(v) - f)$. Then, for every $\epsilon > 0$ there exists an $N \in \mathbb{N}$ such that for all $v, f$ in any compact set $K \subset L^2(\Omega)$, the operator $\mathcal{G}_N = \Pi_N \mathcal{G}(\Pi_N v, \Pi_N f)$ is an $\epsilon$-approximation of $\mathcal{G}(v, f)$, i.e., we have,*

$$\sup_{v,f \in K} \|\mathcal{G}(v, f) - \mathcal{G}_N(v, f)\|_{L^2(\Omega)} \le \epsilon.$$

*Proof.* Note that for an $\epsilon > 0$ there exists an $N = N(\epsilon, d)$ such that for all $v \in K$ we have

$$\sup_{v \in K} \|v - \Pi_N v\|_{L^2(\Omega)} \le \epsilon.$$

Therefore, using the definition of $\mathcal{G}_N$ we can bound the $L^2(\Omega)$ norm of the difference between $\mathcal{G}$ and $\mathcal{G}_N$ as follows,

$$\|\mathcal{G}(v,f) - \Pi_N \mathcal{G}(v_n, f_n)\|_{L^2(\Omega)}$$
$$\leq \|\mathcal{G}(v,f) - \Pi_N \mathcal{G}(v,f)\|_{L^2(\Omega)} + \|\Pi_N \mathcal{G}(v,f) - \Pi_N \mathcal{G}(\Pi_N v, \Pi_N f)\|_{L^2(\Omega)}$$
$$\leq \underbrace{\|\mathcal{G}(v,f) - \Pi_N \mathcal{G}(v,f)\|_{L^2(\Omega)}}_{I} + \underbrace{\|\mathcal{G}(v,f) - \mathcal{G}(\Pi_N v, \Pi_N f)\|_{L^2(\Omega)}}_{II}$$

We first bound the term $I$ as follows:

$$\|\mathcal{G}(v,f) - \Pi_N \mathcal{G}(v,f)\|_{L^2(\Omega)}$$
$$= \|v - (L(v) - f) - \Pi_N (v - (L(v) - f))\|_{L^2(\Omega)}$$
$$= \|v - \Pi_N v\|_{L^2(\Omega)} + \|f - \Pi_N f\|_{L^2(\Omega)} + \|L(v) - \Pi_N L(v)\|_{L^2(\Omega)}$$
$$= \epsilon + \epsilon + \|L(v) - \Pi_N L(v)\|_{L^2(\Omega)} \tag{14}$$

Since $L$ is continuous, for all compact sets $K \subset L^2(\Omega)$, $L(K)$ is compact as well. This is because: (1) for any $u \in K$, $\|L(u)\|_{L^2(\Omega)}$ is finite; (2) for any $v \in K$, $\|L(v)\|_{L^2(\Omega)} \leq \|L(u)\|_{L^2(\Omega)} + C\|u - v\|_{L^2(\Omega)}$. Therefore, for every $\epsilon > 0$, there exists an $N \in \mathbb{N}$ such that

$$\sup_{v \in K} \|L(v) - \Pi_N L(v)\|_{L^2(\Omega)} \leq \epsilon.$$

Substituting the above result in Equation 14, we have

$$\|\mathcal{G}(v,f) - \Pi_N \mathcal{G}(v,f)\|_{L^2(\Omega)} \leq 3\epsilon. \tag{15}$$

Similarly, for all $v \in K$ where $K$ is compact, we can bound Term $II$ as following,

$$\|\mathcal{G}(v,f) - \mathcal{G}(\Pi_N v, \Pi_N f)\|_{L^2(\Omega)}$$
$$\leq \|v - (L(v) - f) - \Pi_N v - (L(\Pi_N v) - \Pi_N f)\|_{L^2(\Omega)}$$
$$\leq \|v - \Pi_N v\|_{L^2(\Omega)} + \|f - \Pi_N f\|_{L^2(\Omega)} + \|L(v) - L(\Pi_N v)\|_{L^2(\Omega)}$$
$$\leq \epsilon + \epsilon + \|L(v) - L(\Pi_N v)\|_{L^2(\Omega)}. \tag{16}$$

Now, since $v \in K$ and $L : L^2(\Omega) \to L^2(\Omega)$ is a continuous operator, there exists a modulus of continuity (an increasing real valued function) $\alpha \in [0, \infty)$, such that for all $v \in K$, we have

$$\|L(v) - L(\Pi_N v)\|_{L^2(\Omega)} \leq \alpha \left( \|v - \Pi_N v\|_{L^2(\Omega)} \right)$$

Hence for every $\epsilon > 0$ there exists an $N \in \mathbb{N}$ such that,

$$\alpha(\|v - \Pi_N v\|_{L^2(\Omega)}) \leq \epsilon.$$

Plugging these bounds in Equation 16, we get,

$$\|\mathcal{G}(v,f) - \mathcal{G}(\Pi_N v, \Pi_N f)\|_{L^2(\Omega)} \leq 3\epsilon. \tag{17}$$

Therefore, combining Equation 15 and Equation 17 then for $\epsilon > 0$, there exists an $N \in \mathbb{N}$, such that for all $v, f \in K$ we have

$$\sup_{v, f \in K} \|\mathcal{G}(v,f) - \Pi_N \mathcal{G}(v_n, f_n)\|_{L^2(\Omega)} \leq 6\epsilon. \tag{18}$$

Taking $\epsilon' = 6\epsilon$ proves the claim. $\qquad \square$

*Proof of Theorem 1.* For Lemma 2 we know that there exists a finite dimensional projection for the operator $\mathcal{G}$, defined as $\mathcal{G}_N(v,f)$ such that for all $v, f \in L^2(\Omega)$ we have

$$\|\mathcal{G}(v,f) - \mathcal{G}_N(v,f)\|_{L^2(\Omega)} \leq \epsilon.$$

Now using the definition of $\mathcal{G}_N(v,f)$ we have

$$\mathcal{G}_N(v,f) = \Pi_N \mathcal{G}(\Pi_N v, \Pi_N f)$$
$$= \Pi_N v - (\Pi_N L(\Pi_N v) - \Pi_N f)$$

From Kovachki et al. [2021a], Theorem 2.4 we know that there exists an FNO network $G_{\theta^L}$ of the form defined in Equation 2 such that for all $v \in K$, where $K$ is a compact set, there exists an $\epsilon^L$ we have

$$\sup_{v \in K} \|\Pi_N L(\Pi_N v) - G_{\theta^L}\|_{L^2(\Omega)} \leq \epsilon^L \tag{19}$$

Finally, note that from Lemma D.1 in Kovachki et al. [2021a], we have that for any $v \in K$, there exists an FNO layers $G_{\theta^f} \in L^2(\Omega)$ and $G_{\theta^v} \in L^2(\Omega)$ defined in Equation 3 such that

$$\sup_{v \in K} \|\Pi_N v - G_{\theta^v}\|_{L^2(\Omega)} \leq \epsilon^v \tag{20}$$

and

$$\sup_{f \in K} \|\Pi_N f - G_{\theta^f}\|_{L^2(\Omega)} \leq \epsilon^f \tag{21}$$

for $\epsilon^v > 0$ and $\epsilon^f > 0$.

Therefore there exists an $\tilde{\epsilon} >$ such that there is an FNO network $G_\theta : L^2(\Omega) \times L^2(\Omega) \to L^2(\Omega)$ where $\theta := \{\theta^L, \theta^v, \theta^f\}$ such that

$$\sup_{v \in K, f \in L^2(\Omega)} \|\mathcal{G}_N(v, f) - G_\theta(v, f)\|_{L^2(\Omega)} \leq \tilde{\epsilon} \tag{22}$$

Now, since we know that $u^\star$ is the fixed point of the operator $\mathcal{G}$ we have from Lemma 2 and Equation 22,

$$\|\mathcal{G}(u^\star, f) - G_\theta(u^\star, f)\|_{L^2(\Omega)} \leq \|u^\star - \mathcal{G}_N(u^\star, f)\|_{L^2(\Omega)} + \|\mathcal{G}_N(u^\star, f) - G_\theta(u^\star, f)\|_{L^2(\Omega)}$$
$$\leq \tilde{\epsilon} + \epsilon.$$

$\square$

# D  Fast Convergence for Newton Method

**Definition 8** (Frechet Derivative in $L^2(\Omega)$). *For a continuous operator $F : L^2(\Omega) \to L^2(\Omega)$, the Frechet derivative at $u \in L^2(\Omega)$ is a linear operator $F'(u) : L^2(\Omega) \to L^2(\Omega)$ such that for all $v \in L^2(\Omega)$ we have*

$$\lim_{\|v\|_{L^2(\Omega)} \to 0} \frac{\|F(u + v) - F(u) - F'(u)(v)\|_{L^2(\Omega)}}{\|v\|_{L^2(\Omega)}} = 0.$$

**Lemma 3.** *Given the operator $L : L^2(\Omega) \to L^2(\Omega)$ with Frechet derivative $L'$, such that for all $u, v \in L^2(\Omega)$, we have $\|L'(u)(v)\|_{L^2(\Omega)} \geq \lambda\|v\|_{L^2(\Omega)}$, then $L'(u)^{-1}$ exists and we have, for all $v_1, v_2 \in L^2(\Omega)$:*

*1. $\|L'(u)^{-1}(v_1)\|_{L^2(\Omega)} \leq \frac{1}{\lambda}\|v_1\|_{L^2(\Omega)}$.*

*2. $\|v_1 - v_2\|_{L^2(\Omega)} \leq \frac{1}{\lambda}\|L(v_1) - L(v_2)\|_{L^2(\Omega)}$*

*Proof.* Note that for all $u, v' \in L^2(\Omega)$ we have,

$$\|L'(u)v'\|_{L^2(\Omega)} \geq \lambda\|v'\|_{L^2(\Omega)}$$

Taking $v = L'(u)^{-1}(v')$, we have

$$\|L'(u)\left(L'(u)^{-1}(v)\right)\|_{L^2(\Omega)} \geq \lambda\|L^{-1}(u)(v)\|_{L^2(\Omega)}$$
$$\implies \frac{1}{\lambda}\|v\|_{L^2(\Omega)} \geq \|L^{-1}(u)(v)\|_{L^2(\Omega)}.$$

For part 2, note that there exists a $c \in [0, 1]$ such that

$$\|L(v_1) - L(v_2)\|_{L^2(\Omega)} \geq \inf_{c \in [0,1]} \|L'(cv_1 + (1-c)v_2)\|_2 \|v_1 - v_2\|_{L^2(\Omega)} \geq \lambda\|v_1 - v_2\|_{L^2(\Omega)}.$$

$\square$

We now show the proof for Lemma 4. The proof is standard and can be found in Faragó and Karátson [2002], however we include the complete proof here for the sake of completeness.

We restate the Lemma here for the convenience of the reader.

**Lemma 4** (Faragó and Karátson [2002], Chapter 5). *Consider the PDE defined Definition 2, such that $d_u = d_v = d_f = 1$. such that $L'(u)$ defines the Frechet derivative of the operator $L$. If for all $u, v \in L^2(\Omega; \mathbb{R})$ we have $\|L'(u)v\|_{L^2(\Omega)} \geq \lambda \|v\|_{L^2(\Omega)}$ [7] and $\|L'(u) - L'(v)\|_{L^2(\Omega)} \leq \Lambda \|u - v\|_{L^2(\Omega)}$ for $0 < \lambda \leq \Lambda < \infty$, then for the Newton update, $u_{t+1} \leftarrow u_t - L'(u_t)^{-1} (L(u_t) - f)$, with $u_0 \in L^2(\Omega; \mathbb{R})$, there exists an $\epsilon > 0$, such that $\|u_T - u^\star\|_{L^2(\Omega)} \leq \epsilon$ if [8] $T \geq \log \left( \log \left( \frac{1}{\epsilon} \right) / \log \left( \frac{2\lambda^2}{\Lambda \|L(u_0) - f\|_{L^2(\Omega)}} \right) \right).$*

*Proof of Lemma 4.* Re-writing the updates in Lemma 4 as,

$$u_{t+1} = u_t + p_t \tag{23}$$
$$L'(u_t)p_t = -(L(u_t) - f) \tag{24}$$

Now, upper bounding $L(u_{t+1}) - f$ for all $x \in \Omega$ we have,

$L(u_{t+1}(x)) - f(x)$

$$= L(u_t(x)) - f(x) + \int_0^1 \left( L'(u_t(x) + t(u_{t+1}(x) - u_t(x))) \right) (u_{t+1}(x) - u_t(x)) dt$$

$$= L(u_t(x)) - f(x) + L'(u_t(x))p_t(x) + \int_0^1 \left( L'(u_t(x) + t(u_{t+1}(x) - u_t(x))) - L'(u_t(x)) \right) p_t(x) dt$$

$$= \int_0^1 \left( L'(u_t(x) + t(u_{t+1}(x) - u_t(x))) - L'(u_t(x)) \right) p_t(x) dt$$

where we use Equation 24 in the final step.

Taking $L^2(\Omega)$ norm on both sides and using the fact that $\|L'(u) - L'(v)\|_{L^2(\Omega)} \leq \Lambda \|u - v\|_{L^2(\Omega)}$, we have

$$\|L(u_{t+1}) - f\|_{L^2(\Omega)} \leq \int_0^1 \Lambda t \|u_{t+1} - u_t\|_{L^2(\Omega)} \|p_t\|_{L^2(\Omega)} dt$$

Noting that for all $x \in \Omega$, we have $u_{t+1} - u_t = p_t$, and using the fact that for all $u, v$ $\|L'(u)^{-1}v\|_{L^2(\Omega)} \leq \frac{1}{\lambda} \|v\|_{L^2(\Omega)}$ we have, $\|L'(u_t)p_t\|_{L^2(\Omega)} \leq \frac{1}{\lambda} \|p_t\|_{L^2(\Omega)}$

$$\|L(u_{t+1}) - f\|_{L^2(\Omega)} \leq \int_0^1 \Lambda t \|u_{t+1} - u\|_{L^2(\Omega)} \|p_t\|_{L^2(\Omega)} dt$$
$$\leq \Lambda/2 \|p_t\|_{L^2(\Omega)}^2$$
$$\leq \Lambda/2 \| - L'(u_t)^{-1}(L(u_t) - f)\|_{L^2(\Omega)}^2$$
$$\leq \frac{\Lambda}{2\lambda^2} \|L(u_t) - f)\|_{L^2(\Omega)}^2$$

where we use the result from Lemma 3 in the last step.

Therefore we have

$$\|L(u_{t+1}) - f\|_{L^2(\Omega)} \leq \left( \frac{\Lambda}{2\lambda^2} \right)^{2^t - 1} (L(u_0) - f)^{2^t}$$

$$\implies \|L(u_{t+1}) - f\|_{L^2(\Omega)} \leq \left( \frac{\Lambda}{2\lambda^2} \right)^{2^t - 1} (L(u_0) - L(u^\star))^{2^t}$$

$$\implies \|u_{t+1} - u^\star\|_{L^2(\Omega)} \leq \frac{1}{\lambda} \left( \frac{\Lambda}{2\lambda^2} \right)^{2^t - 1} \|L(u_0) - L(u^\star)\|_{L^2(\Omega)}^{2^t}.$$

---

[7] We note that this condition is different from the condition on the inner-product in the submitted version of the paper, which had. $\langle L'(u), v \rangle_{L^2(\Omega)} \geq \lambda \|v\|_{L^2(\Omega)}$.

[8] We note that this rate is different from the one in the submitted version of the paper.

Therefore, if

$$\frac{\Lambda}{2\lambda^2}\|L(u_0) - L(u^\star)\|_{L^2(\Omega)} \leq 1,$$

then we have

$$\|u_{t+1} - u^\star\|_{L^2(\Omega)} \leq \epsilon,$$

for

$$T \geq \log\left(\log\left(\frac{1}{\epsilon}\right) \Big/ \log\left(\frac{2\lambda^2}{\Lambda\|L(u_0) - f\|_{L^2(\Omega)}}\right)\right).$$

$\square$

# E   Additional experimental results

We provide additional results for Navier-Stokes equation for noisy inputs and observations in Table 4 and Table 5. For these experiments, the maximum variance of Gaussian noise added to inputs and observations is 0.004. We observe that weight-tied FNO and FNO-DEQ outperform non-weight-tied architectures.

| Architecture | Parameters | #Blocks | Test error ↓ | | |
|---|---|---|---|---|---|
| | | | $\sigma_{max}^2 = 0$ | $(\sigma_{max}^2)^i = 0.004$ | $(\sigma_{max}^2)^t = 0.004$ |
| FNO | 2.37M | 1 | $0.184 \pm 0.002$ | $0.238 \pm 0.008$ | $0.179 \pm 0.004$ |
| FNO | 4.15M | 2 | $0.162 \pm 0.024$ | $0.196 \pm 0.011$ | $0.151 \pm 0.010$ |
| FNO | 7.71M | 4 | $0.157 \pm 0.012$ | $0.216 \pm 0.002$ | $0.158 \pm 0.009$ |
| FNO++ | 2.37M | 1 | $0.199 \pm 0.001$ | $0.255 \pm 0.002$ | $0.197 \pm 0.004$ |
| FNO++ | 4.15M | 2 | $0.154 \pm 0.005$ | $0.188 \pm 0.006$ | $0.157 \pm 0.006$ |
| FNO++ | 7.71M | 4 | $0.151 \pm 0.003$ | $0.184 \pm 0.008$ | $0.147 \pm 0.004$ |
| FNO-WT | 2.37M | 1 | $0.151 \pm 0.007$ | $0.183 \pm 0.026$ | $0.129 \pm 0.018$ |
| FNO-DEQ | 2.37M | 1 | $\mathbf{0.128 \pm 0.004}$ | $\mathbf{0.159 \pm 0.005}$ | $\mathbf{0.121 \pm 0.015}$ |

Table 4: **Results on incompressible Steady-State Navier-Stokes (viscosity=0.001)**: clean data (Col 4), noisy inputs (Col 5) and noisy observations (Col 6) with max variance of added noise being $(\sigma_{max}^2)^i$ and $(\sigma_{max}^2)^t$, respectively. Reported test error has been averaged on three different runs with seeds 0, 1, and 2.
‡ indicates that the network diverges during training for one of the seeds.

| Architecture | Parameters | #Blocks | Test error ↓ | | |
|---|---|---|---|---|---|
| | | | $\sigma_{max}^2 = 0$ | $(\sigma_{max}^2)^i = 0.004$ | $(\sigma_{max}^2)^t = 0.004$ |
| FNO | 2.37M | 1 | $0.181 \pm 0.005$ | $0.207 \pm 0.003$ | $0.178 \pm 0.008$ |
| FNO | 4.15M | 2 | $0.138 \pm 0.007$ | $0.163 \pm 0.003$ | $0.137 \pm 0.006$ |
| FNO | 7.71M | 4 | $0.152 \pm 0.006$ | $0.203 \pm 0.055$ | $0.151 \pm 0.008$ |
| FNO++ | 2.37M | 1 | $0.188 \pm 0.002$ | $0.217 \pm 0.001$ | $0.187 \pm 0.005$ |
| FNO++ | 4.15M | 2 | $0.139 \pm 0.004$ | $0.170 \pm 0.005$ | $0.138 \pm 0.005$ |
| FNO++ | 7.71M | 4 | $0.130 \pm 0.005$ | $0.168 \pm 0.007$ | $0.126 \pm 0.007$ |
| FNO-WT | 2.37M | 1 | $0.099 \pm 0.007$ | $0.159 \pm 0.029$ | $0.123 \pm 0.023$ |
| FNO-DEQ | 2.37M | 1 | $\mathbf{0.088 \pm 0.006}$ | $\mathbf{0.104 \pm 0.001}$ | $\mathbf{0.116 \pm 0.005}$ |

Table 5: **Results on incompressible Steady-State Navier-Stokes (viscosity=0.01)**: clean data (Col 4), noisy inputs (Col 5) and noisy observations (Col 6) with max variance of added noise being $(\sigma_{max}^2)^i$ and $(\sigma_{max}^2)^t$, respectively. Reported test error has been averaged on three different runs with seeds 0, 1, and 2.
‡ indicates that the network diverges during training for one of the seeds.

**Convergence analysis of fixed point.** We report variations in test error, absolute residual $\|G_\theta(\mathbf{z}_t) - \mathbf{z}_t\|_2$, and relative residual $\frac{\|G_\theta(\mathbf{z}_t) - \mathbf{z}_t\|_2}{\|\mathbf{z}_t\|_2}$ with an increase in the number of solver steps while solving for the fixed point in FNO-DEQ, for both Darcy Flow (See Table 6) and Steady-State Navier Stokes (See Table 7). We observe that all these values decrease with increase in the number of fixed point solver iterations and eventually saturate once we have a reasonable estimate of the fixed point. We observe that increasing the number of fixed point solver iterations results in a better estimation of the fixed point. For steady state PDEs, we expect the test error to reduce as the estimation of the fixed point improves. Furthermore, at inference time we observe that the test error improves (i.e. reduces) with increase in the number of fixed point solver iterations even though the FNO-DEQ is trained with fewer solver steps. For Navier-Stokes with viscosity 0.01, at inference time we get a test MSE loss of 0.0744 with 48 solver steps from 0.0847 when used with 24 solver steps.

This further bolsters the benefits of DEQs (and weight-tied architectures in general) for training neural operators for steady-state PDEs. Moreover, performance saturates after a certain point once we have a reasonable estimate of the fixed point, hence showing that more solver steps stabilize to the same solution.

| Solver steps | Absolute residual ↓ | Relative residual ↓ | Test Error ↓ |
|---|---|---|---|
| 2 | 212.86 | 0.8533 | 0.0777 |
| 4 | 18.166 | 0.0878 | 0.0269 |
| 8 | 0.3530 | 0.00166 | 0.00567 |
| 16 | 0.00239 | 1.13e-5 | 0.00566 |
| 32 | 0.000234 | 1.1e-6 | 0.00566 |

Table 6: Convergence analysis of fixed point for noiseless Darcy Flow: The test error, absolute residual $\|G_\theta(\mathbf{z}_t) - \mathbf{z}_t\|_2$ and relative residual $\frac{\|G_\theta(\mathbf{z}_t) - \mathbf{z}_t\|_2}{\|\mathbf{z}_t\|_2}$ decrease with increase in the number of fixed point solver iterations. The performance saturates after a certain point once we have a reasonable estimate of the fixed point. We consider the noiseless case, where we do not add any noise to inputs or targets.

| Solver steps | Absolute residual ↓ | Relative residual ↓ | Test Error ↓ |
|---|---|---|---|
| 4 | 544.16 | 0.542 | 0.926 |
| 8 | 397.75 | 0.408 | 0.515 |
| 16 | 150.33 | 0.157 | 0.147 |
| 24 | 37.671 | 0.0396 | 0.0847 |
| 48 | 5.625 | 0.0059 | 0.0744 |
| 64 | 3.3 | 0.0034 | 0.0746 |

Table 7: Convergence analysis of fixed point for noiseless incompressible Steady-State Navier-Stokes with viscosity=0.01: The test error, absolute residual $\|G_\theta(\mathbf{z}_t) - \mathbf{z}_t\|_2$ and relative residual $\frac{\|G_\theta(\mathbf{z}_t) - \mathbf{z}_t\|_2}{\|\mathbf{z}_t\|_2}$ decrease with increase in the number of fixed point solver iterations. The performance saturates after a certain point once we have a reasonable estimate of the fixed point. We consider the noiseless case, where we do not add any noise to inputs or targets.

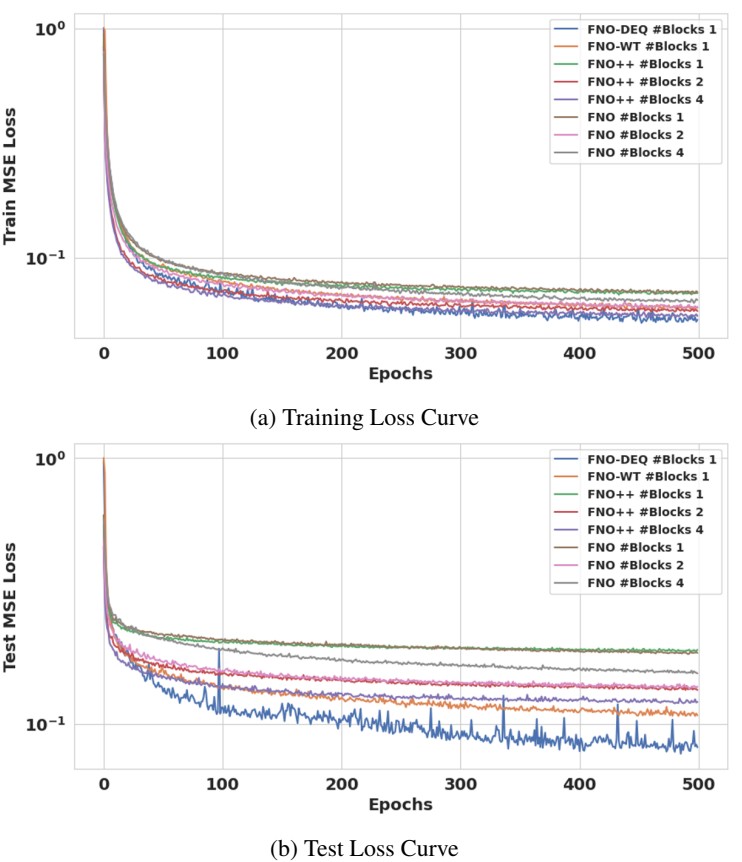

(a) Training Loss Curve

(b) Test Loss Curve

Figure 4: Training and Test Loss Curves for Steady-State Navier-Stokes with viscosity $0.01$. The $x$ axis is the number of epochs and $y$ axis is the MSE loss in $\log$ scale. Note that while all the models converge to approximately the same MSE loss value while training, DEQs and weight-tied networks get a better test loss in fewer epochs.

