# OpenReview forum: "Deep Equilibrium Based Neural Operators for Steady-State PDEs"
_NeurIPS.cc/2023/Conference — NeurIPS 2023 poster_

### Official Review · Reviewer_SM2z · 2023-06-16

**Soundness:** 4 excellent
**Presentation:** 4 excellent
**Contribution:** 4 excellent
**Rating:** 6
**Confidence:** 5

**Summary:**

The paper proposes a method combining Fourier Neural Operator (FNO) with the Deep Equilibrium Model (DEQ) for more efficient operator learning under noisy conditions. Overall, the paper is well-written with clear methods and theory, and some effective experimental results. It's a decent piece of work, and I recommend a weak acceptance. However, I believe there are still areas in the paper that could be improved.

**Strengths:**

1. The paper is well-written, with a clear presentation of the method and theoretical underpinnings.

2. The experiments show some positive results, demonstrating the potential utility of the proposed approach.



**Weaknesses:**

1. The paper lacks a thorough literature review. It hardly mentions operator learning methods based on attention, which have demonstrated significant advantages in many areas, such as Navier-Stokes equations and problems on irregular geometric regions. Therefore, I believe the authors should add some references [1,2,3] to cover these works.

2. Each experiment in the paper compares the effects of adding noise to either the input or output. However, the improvements observed in noisy settings do not seem to be as significant as those without noise. This makes me wonder why the authors considered these experiments necessary.

3. Additionally, as a new model structure, DEQ should be able to be combined with many other approaches, such as DeepONet or other neural operators. However, the paper seems not to mention this possibility and only tries to combine it with FNO. It would be interesting to see an exploration of the potential of DEQ in combination with other models.

References
1. Transformer for Partial Differential Equations' Operator Learning (https://arxiv.org/abs/2205.13671)

2. GNOT: A General Neural Operator Transformer for Operator Learning (https://arxiv.org/abs/2302.14376)

3. HT-Net: Hierarchical Transformer based Operator Learning Model for Multiscale PDEs (https://arxiv.org/abs/2210.10890)


**Questions:**

None

**Limitations:**

None.

---

> ### Author Rebuttal · Authors · 2023-08-10
>
> We thank the reviewer for their feedback and are encouraged to see that the reviewer finds the paper to be well-written with positive experimental results.
>
> Please find our replies to some of your comments and concerns:
>
> **Re: Lacking references to attention based operator learning.**
> We apologize for not including the mentioned papers in our literature review and will be sure to include them in the Related Work section in the revised manuscript.
>
> **Re: Importance of experiments with noise.**
> The primary motivation behind showing results with added noise was to emulate real-world scenarios where the different physical quantities in a PDE are measured using sensors—since they are likely to incur noise in the observations.
> We further note that the relative decrease in the MSE loss for weight-tied architectures (FNO-DEQ and FNO-WT) is smaller than when compared to FNO and FNO++ especially when trained with noisy inputs. For example, for Navier-Stokes with viscosity 0.01 we only see a 7% decrease in the performance of FNO-DEQ (best performing weight-tied model) versus 23% decrease in best performing FNO++ model.
> Therefore, weight tied architectures are indeed an effective inductive bias for solving steady-state PDEs with neural operator frameworks.
>
>
> **Re: Weight-tied DeepONets**
> In principle, weight-tieing and DEQs can be combined with other neural operators like DeepONets. We chose FNO in part due its performance on various benchmark tasks in PDEbench [1]. We certainly hope that experiments with other operator architectures will be performed, by us and the community, and think this is fertile ground for future work!
>
>
> [1] Takamoto, Makoto, et al. "PDEBench: An extensive benchmark for scientific machine learning." Advances in Neural Information Processing Systems 35 (2022): 1596-1611.

---

> > ### Author Response · Authors · 2023-08-21
> >
> > Dear reviewer, thank you for your review and feedback! Since today is the last day of the author-reviewer discussion period, please let us know if there are any outstanding questions that we can help answer or clarify.

---

### Official Review · Reviewer_Kt2x · 2023-07-06

**Soundness:** 2 fair
**Presentation:** 3 good
**Contribution:** 3 good
**Rating:** 5
**Confidence:** 1

**Summary:**

The paper proposes two weight-tied neural network architectures for solving steady-state partial differential equations (PDEs) using the universal approximation capabilities of neural networks. The first architecture is a weight-tied version of Fourier Neural Operators (FNO), while the second architecture is a Deep Equilibrium Model (DEQ) that uses black-box root finding algorithms to implicitly train the model. The paper shows that both architectures outperform existing methods on benchmark problems and can be used to learn efficient solvers for PDEs. The contributions of the paper include the introduction of the weight-tied FNO and FNO-DEQ architectures, as well as the demonstration of their effectiveness in solving steady-state PDEs.

**Strengths:**

The main strengths of the paper are:
1. The paper proposes a new architecture called FNO-DEQ that uses weight-tied neural network layers to solve steady-state partial differential equations (PDEs). The architecture is based on the observation that the solution of most steady-state PDEs can be expressed as a fixed point of a non-linear operator.
2. The paper shows that FNO-DEQ outperforms other non-weight-tied architectures with 4x the number of parameters in predicting the solution to steady-state PDEs such as Darcy Flow and steady-state incompressible Navier-Stokes.
3. The paper demonstrates that FNO-DEQ and weight-tied architectures are more robust to both input and observation noise compared to non-weight-tied architectures, including FNO.
4. The paper leverages the universal approximation results of FNO to show that FNO-DEQ can universally approximate the solution operator for a wide variety of steady-state PDE families.

**Weaknesses:**

Some potential limitations of the paper are:
1. The proposed FNO-DEQ architecture may not be applicable to all types of PDEs, as it is specifically designed for steady-state PDEs. Further research is needed to explore the effectiveness of weight-tied architectures for other types of PDEs.
2. The paper focuses on the empirical performance of the proposed approach and does not provide a detailed theoretical analysis of why weight-tying is effective for steady-state PDEs. A more rigorous theoretical analysis could provide deeper insights into the underlying mechanisms of the proposed approach.
3. The paper does not compare the proposed approach to other state-of-the-art methods for solving steady-state PDEs, such as finite element methods or spectral methods. A more comprehensive comparison could provide a better understanding of the relative strengths and weaknesses of different approaches.
4. The proposed approach may require more computational resources than other methods, as it involves solving for the fixed point of an implicit operator layer. This could limit its scalability to larger or more complex PDEs.

**Questions:**

Refer to the weaknesses section.

**Limitations:**

Refer to the weaknesses section.

---

> ### Author Rebuttal · Authors · 2023-08-10
>
> We thank the reviewer for your positive review! Please find our responses to the concerns you raised below.
>
> **Re: Proposed architectures not applicable to all PDEs.**
> There are a multitude of different forms of PDEs, each with their unique characteristics that may or may not be efficiently modeled by a single neural network architecture. In our paper we show that for a *broad class of PDEs*, namely *steady state PDEs*, there are specific architectural choices that can outperform the baselines with much fewer parameters. We fully agree that charting the space of effective architectures for various PDE families is a major outstanding question in using machine learning methods for PDE solvers.
>
> **Re: Lack of theoretical benefits of why weight-tying is beneficial for steady-state PDEs.**
> We agree that more theoretical understanding of the benefits of different kinds of architectures is an exciting avenue for further research, and we hope that there is a concerted effort towards studying and designing architectures for PDEs by the machine learning community!
> However, we would like to note that in general showing separations theoretically between the performance (statistical or algorithmic) of different architectures is incredibly challenging to show theoretically—e.g. sample complexity separations between convnets and fully connected networks are very poorly understood, and most of the results are under very strong assumptions [4].
>
> That being said, as we mention in our paper, there indeed are some theoretical motivations towards using weight-tied architectures for steady-state PDEs as established in the previous works of [1,2,3] which we use as a motivation for our work.
>
> **Re: Comparison with finite element method and spectral methods.**
> Our paper builds on neural operators (in particular Fourier Neural Operators (FNO) [5]), which have many benefits over fixed algorithms like finite element methods or spectral methods. Some of the benefits are:
> - **Computational efficiency**: Neural operators learn a solution operator for an entire family of PDE, which has an inference time of just 0.005s compared to the 2.2s of spectral methods when used to solve Navier-Stokes equations [5]. Similar gains can be observed for many families of PDEs.
> - **Robustness to discretization**: Neural operators (especially FNO) are robust to changes in discretization at inference time, that is they tend to perform well even if some of the training data is very coarsely discretized.
>
> We refer the reviewer to [5] and related works for a more comprehensive overview of the benefits of neural operators.
>
> **Re: Scaling to more complex PDEs.**
> It is true that the computational overhead would increase with more complex PDEs, however that is true for *all* neural network based PDE solvers and is not limited to FNO-DEQ or FNO-WT. In fact, we believe weight-tied and DEQ based architectures are better suited for more complex steady-state PDEs since the memory used by DEQs for the backward pass is going to be constant, i.e., O(1),  but for deeper architectures in FNO and FNO++,  the memory required would increase with the number of layers in the architecture.
>
> However, as mentioned in our paper, training DEQs can be slow as we solve for a fixed point in the forward pass. Through use of approximate implicit gradients and with careful selection of hyperparameters, the training can be made faster. Further, when compared to a non-weight-tied network of a similar depth, the overhead due to fixed point solving is marginal, both in terms of compute and memory requirements.
>
> [1] Marwah, Tanya, et al. "Neural Network Approximations of PDEs Beyond Linearity: A Representational Perspective." International Conference on Machine Learning. PMLR, 2023.
>
> [2] Marwah, Tanya, Zachary Lipton, and Andrej Risteski. "Parametric complexity bounds for approximating PDEs with neural networks." Advances in Neural Information Processing Systems 34 (2021): 15044-15055.
>
> [3] Chen, Ziang, Jianfeng Lu, and Yulong Lu. "On the representation of solutions to elliptic pdes in barron spaces." Advances in neural information processing systems 34 (2021): 6454-6465.
>
> [4] Li, Zhiyuan, Yi Zhang, and Sanjeev Arora. "Why are convolutional nets more sample-efficient than fully-connected nets?." arXiv preprint arXiv:2010.08515 (2020).
>
> [5] Li, Zongyi, et al. "Fourier neural operator for parametric partial differential equations." arXiv preprint arXiv:2010.08895 (2020)

---

> > ### Author Response · Authors · 2023-08-21
> >
> > Dear reviewer, thank you for your review and feedback! Since today is the last day of the author-reviewer discussion period, please let us know if there are any outstanding questions that we can help answer or clarify.

---

### Official Review · Reviewer_wRdk · 2023-07-06

**Soundness:** 4 excellent
**Presentation:** 3 good
**Contribution:** 4 excellent
**Rating:** 9
**Confidence:** 5

**Summary:**

This research examined the solution of steady-state Partial Differential Equations (PDEs) using Fourier Neural Operator (FNO) based architecture. The author introduced a fix-point iteration mechanism into the FNO framework, leading to the proposal of weight-tied FNO and FNO Deep Equilibrium (FNO-DEQ) models. Comparative analyses revealed that these newly proposed architectures outperformed the traditional FNO when solving standard Darcy Flow and Navier-Stokes Equations.

**Strengths:**

This paper is well written, providing clear and rigorous mathematical definitions of the problem, its underlying theory, and the proposed solution.

The innovation of incorporating the fixed-point iteration mechanism (via the contraction mapping theorem) into the Fourier Neural Operator (FNO) is novel and captivating. The results clearly demonstrate a significant improvement when applying this technique to steady-state Partial Differential Equations (PDEs).

Further enhancing the strength of the paper, the author proves a universal approximation theorem for the FNO Deep Equilibrium (FNO-DEQ) model, which assures the boundedness of the approximation. This theoretical validation lends additional credibility and rigor to the findings.


**Weaknesses:**

The paper doesn't provide any loss versus training epochs data, which would allow us to assess whether the training had indeed reached convergence (as well as to compare the speed of convergence). While it's assumed that convergence must have been achieved for the results shown in Tables 2 and 3, the absence of this specific data inhibits a more comprehensive understanding of the model's training process.

**Questions:**

The paper doesn't provide a clear explanation regarding the practical implementation of the Fourier Neural Operator Deep Equilibrium (FNO-DEQ) model. The inclusion of a brief paragraph or section detailing its implementation would greatly enhance the reader's understanding and potentially facilitate the model's broader adoption.

**Limitations:**

The author focused only on the standard Darcy flow and Navier Stokes problem, which the classical FNO already works well. An immediete question is that if the FNO-DEQ can be applied to more challenging PDEs (for example 3D Navier Stokes), to show its effectiveness, where classical FNO struggles.

---

> ### Author Rebuttal · Authors · 2023-08-10
>
> Thanks for the encouraging review and feedback. We are glad that the reviewer finds our paper as novel and captivating!!  Please find our replies to some of your comments below. We promise to make the corresponding changes to the camera-ready version of our draft to incorporate your suggestions.
>
> **Re: Loss vs training epochs data and convergence.**
> Thank you for your feedback! We agree that adding convergence plots will help towards a more comprehensive understanding of our methodology. We have added the convergence plots for Naiver Stokes with viscosity 0.01 in the attached PDF (Figure 1). The key observation is that while all the models converge to approximately the same train MSE value, the convergence of the test loss differs, i.e., DEQs and weight tied networks get a better test loss in fewer epochs.
> We also note that the convergence plots for Darcy Flow and Navier-Stokes with viscosity 0.001 follow similar trends. We will add the convergence plots for all the PDEs and models in the final version of the paper.
>
> **Re: Practical implementation and clarification.**
> We provide a detailed description of the implementation and architectural details in the Appendix (Section A) of the supplementary material. For example, we note that all the networks for 500 epochs with Adam optimizer with L2 weight penalty coefficient of 1e-4. The learning rate is set to 0.001 for Darcy flow and 0.005 for Navier-Stokes, with batch size 32. Further information about the training details of the DEQ architecture are provided in the Appendix (Section A) as well. We will restructure the draft to include the key implementation details earlier in the paper.
>
> Finally, we will release our code (submitted as supplementary material) detailing the implementation details and appropriate documentation on how to reproduce our results along with the camera-ready version of the paper.
>
> **Re: Standard Darcy Flow and Navier-Stokes problem where FNO already works well.**
> We agree that applying FNO-DEQ and FNO-WT to more challenging PDEs is an important step towards showcasing the benefits of the proposed architectures, and we believe scaling our methodologies to more complex PDEs is a fertile ground for future work. However, we also would like to note that ours is the first paper to benchmark the performance of FNO (and FNO++) along with weight-tied models on steady-state 2D Navier-Stokes PDEs!

---

> > ### Comment · Reviewer_wRdk · 2023-08-20
> >
> > Thanks for the rebuttal. All my questions are well answered.

---

### Official Review · Reviewer_HGbF · 2023-07-13

**Soundness:** 3 good
**Presentation:** 3 good
**Contribution:** 3 good
**Rating:** 7
**Confidence:** 3

**Summary:**

The paper tackles the problem of solving steady-state PDEs with weight-tying FNOs. The authors argue that instead of stack multiple FNO layers with different parameters, repeatedly applying one FNO layer computation is a better choice. This hypothesis is motivated by the fact that steady-state PDEs are solved for fixed-point solutions, where evolving PDE further will not change the solution. Moreover, instead of directly stack the same FNO layer and hope the fix point can be reached, the authors propose to use more advanced fixed-point solving method (FNO-DEQ). By using the fix-point properties, DEQ can have smaller training memory usage but is slower. Experiments are conducted on Darcy flow and naiver-stokes equations.

**Strengths:**

- The paper is well structured,
- The proposed method leverages well the fact that the solutions are fixed points.
- Experimental results look good.

**Weaknesses:**

- There are very few discussions on the convergence of the fixed point solution. For example, if we apply the FNO layers more times than during training, would the solution remain the same?
- There are two major differences between FNO-WT and FNO-DEQ. The first is in forward pass FNO-WT directly applies the network multiple times while FNO-DEQ use fixed point solver. The second is in the backward pass FNO-WT directly propagates through the computation graph while FNO-DEQ use implicit gradients. These two components seem independent to each other. For example, since FNO-WT in some sense also solve for the fixed point solution, can we use the implicit gradient to train FNO-WT?

**Questions:**

- How is the speed of computing the implicit gradient?
- In Definition 3, shouldn't all functions map from $\Omega$?

---

> ### Author Rebuttal · Authors · 2023-08-10
>
> We thank the reviewer for their positive feedback and response! We are glad that the reviewer finds that our paper is well structured and likes the empirical results. We address some of the concerns raised by the reviewer below:
>
> **Re: A discussion on the convergence of the fixed point.**
> Increasing the number of fixed point solver iterations results in a better estimation of the fixed point. For steady state PDEs, we expect the test error to reduce as the estimation of the fixed point improves. Furthermore, at inference time we observe that the test error improves (i.e. reduces) with increase in the number of fixed point solver iterations even though the DEQ is trained with fewer solver steps.
>
> We report these empirical results for Darcy flow and steady state Navier Stokes (viscosity 0.01) in Tables 1 and 2 in the attached PDF. For example, for Navier-Stokes with viscosity 0.01, at inference time we get a test MSE loss of 0.0744 with 48 solver steps from 0.0847 when used with 24 solver steps.
>
> This further bolsters the benefits of DEQs (and weight-tied architectures in general) for training neural operators for steady-state PDEs. Moreover, performance saturates after a certain point once we have a reasonable estimate of the fixed point, hence showing that more solver steps stabilize to the same solution.
>
> While we only show convergence graphs for the noiseless experiments due to space constraints, we will include similar results for all the PDEs as well as experiments with noise in the final version of the paper.
>
> **Re: Difference between FNO-WT and FNO-DEQ.**
> Both FNO-WT and FNO-DEQ leverage weight-tying (i.e., applying the same transformation at each layer) as the fundamental architectural choice. Where they differ is at the final aim of the forward pass: while FNO-WT will have a *fixed depth* computation (that may or may not approach a fixed point), FNO-DEQ explicitly is trained to find/tends to a fixed point. Therefore, since FNO-WT might fail to converge to a fixed point, implicit gradients cannot be used for FNO-WT.
>
> We refer the reviewer to Bai et al. (2019) for further details on the difference between equilibrium models and traditional weight-tied models that repeat a transformation for a fixed depth.
>
> **Re: Speed of computing implicit gradients**
> We use phantom gradients [2] to compute approximate implicit gradients. These gradients are very light-weight, and fast to compute as they require backward pass through a single FNO block only a couple of times (1 time for Darcy Flow, and 3 times for Navier Stokes). More details of phantom gradients are included in Line 455-459 in the Supplementary Material.
>
> **Re: Clarification on the domains of functions.**
> We note that since the projection operator $\mathcal{P}$ is a function from $\Omega$ to $\mathbb{R}^d$, the domain for all the other functions in the composition would be $\mathbb{R}^d$. We will ensure that this is more clear in the final version of the paper.
>
>
>
> [1] Bai, Shaojie, J. Zico Kolter, and Vladlen Koltun. "Deep equilibrium models." Advances in Neural Information Processing Systems 32 (2019).
>
> [2] Geng, Zhengyang, et al. "On training implicit models." Advances in Neural Information Processing Systems 34 (2021): 24247-24260.

---

> > ### Comment · Reviewer_HGbF · 2023-08-15
> >
> > Thanks for the authors' response. While deep equilibrium model is new to me, I believe this is an interesting paper. I feel some theoretical analysis on the convergence would improve the presentation but I think this should be optional. I will increase my score.

---

### Author Rebuttal · Authors · 2023-08-10

We thank the reviewers for their feedback and the detailed comments, and are encouraged to see the overall positive response for our paper! We hope that we have sufficiently addressed all the questions and comments posed by the reviewers in their individual responses.

Furthermore, we have also attached a PDF with relevant figures and numbers that accompany our individual replies to all the reviewers.

---

### Decision · Program_Chairs · 2023-09-21

**Decision:**

Accept (poster)

**Comment:**

All reviewers agree that the paper brings new insights and significance to the problem by investigating solutions of steady-state PDEs using Fourier Neural Operator (FNO). Also during the discussion phase, the authors were able to provide useful feedback to the comments given by the reviewers, which was helpful in addressing their technical concerns. As a whole, I recommend acceptance of the paper for this submission.